# Data Redaction from Pre-trained GANs

**Zhifeng Kong**
University of California San Diego
La Jolla, CA 92093
z4kong@eng.ucsd.edu

**Kamalika Chaudhuri**
University of California San Diego
La Jolla, CA 92093
kamalika@eng.ucsd.edu

## Abstract

Large pre-trained generative models are known to occasionally output undesirable samples, which undermines their trustworthiness. The common way to mitigate this is to re-train them differently from scratch using different data or different regularization – which uses a lot of computational resources and does not always fully address the problem. In this work, we take a different, more compute-friendly approach and investigate how to post-edit a model after training so that it "redacts", or refrains from outputting certain kinds of samples. We show that redaction is different from data deletion, and data deletion may not always lead to redaction. We then consider Generative Adversarial Networks (GANs), and provide three different algorithms for data redaction that differ on how the samples to be redacted are described. Extensive evaluations on real-world image datasets show that our algorithms out-perform data deletion baselines, and are capable of redacting data while retaining high generation quality at a fraction of the cost of full re-training.

## 1   Introduction

Generative Adversarial Networks (GANs) are large neural generative models that learn a complicated probability distribution from data and then generate samples from it. These models have been immensely successful in many large scale tasks from multiple domains, such as images [Zhu et al., 2020, Karras et al., 2020, 2021], point clouds [Zhang et al., 2021], video [Tulyakov et al., 2018], text [de Masson d'Autume et al., 2019], and speech [Kong et al., 2020].

However, it is also well-known that many deep generative models frequently output undesirable samples, which makes them less reliable and trustworthy. Image models generate blurred samples [Kaneko and Harada, 2021] or checkerboard artifacts [Odena et al., 2016, Zhang et al., 2019, Wang et al., 2020, Schwarz et al., 2021], speech models produce unnatural sound [Donahue et al., 2018, Thiem et al., 2020], and language models emit offensive text [Abid et al., 2021, Perez et al., 2022]. Thus, an important question is how to mitigate these artifacts, which would improve the trustworthiness of these models.

One way to mitigate undesirable samples is to re-design the entire training pipeline including data augmentation, model architecture and loss functions, and then re-train the entire model from scratch [Isola et al., 2017, Aitken et al., 2017, Kaneko and Harada, 2021] – a strategy that has been used in prior work. This approach is very compute-intensive as modern GANs can be extremely expensive to train. In addition, other problems may become apparent after training, and resolving them may require multiple re-trainings. To address this challenge, we consider *post-editing*, which means modifying a pre-trained model in a certain way rather than training it differently from scratch. This is a much more computationally efficient process that has shown empirical success in many supervised learning tasks [Frankle and Carbin, 2018, Zhou et al., 2021, Taha et al., 2021], but has not been studied much for unsupervised learning. In particular, we propose a post-editing framework to redact undesirable samples that might be generated by a GAN, which we call *data redaction*.

2022 Trustworthy and Socially Responsible Machine Learning (TSRML 2022) co-located with NeurIPS 2022.

A second plausible solution for mitigating undesirable samples is to use a classifier to filter them out after generation. This approach, however, has several drawbacks. Classifiers can take a significant amount of space and time after deployment. Additionally, if the generative model is handed to a third party, then the model trainer has no control over whether the filter will ultimately be used. Data redaction via post-editing, on the other hand, offers a cleaner solution which does not suffer from these limitations.

A third plausible solution is data deletion or machine unlearning – post-edit the model to approximate a re-trained model that is obtained by re-training from scratch after removing the undesirable samples from the training data. However, this does not always work – as we show in Section E.3, deletion does not necessarily lead to redaction in constrained models. Additionally, the undesirable samples may simply be artifacts of the neural generative model and may not exist in the training data; examples include unnatural sounds emitted by speech models and blurred images from image models. Data redaction, in contrast, can address all these challenges.

There are two major technical challenges that we need to resolve in order to do effective data redaction. The first is how to describe the samples to be redacted. This is important as data redaction algorithms need to be tailored to specific descriptions. The second challenge is that we need to carefully balance data redaction with retaining good generation quality, which means the latent space and the networks must be carefully manipulated.

In this work, we propose a systematic framework for redacting data from pre-trained generative models (see Section 2). We model data redaction as learning the data distribution restricted to the complement of a redaction set $\Omega$. We then formalize three ways of describing redaction sets, namely data-based (where a pre-specified set is given), validity-based (where there is a validity checker), and classifier-based (where there is a differentiable classifier).

Then, we introduce three data redaction algorithms, one for each description (see Section 3). Prior works have looked at avoiding negative samples in the re-training setting with different descriptions and purposes [Sinha et al., 2020, Asokan and Seelamantula, 2020]. They introduce fake distributions to penalize the generation of negative samples. We extend this idea to data redaction by defining the fake distribution as a mixture of the generative distribution and a redaction distribution supported on $\Omega$. We prove the optimal generator can recover the target distribution when label smoothing [Salimans et al., 2016, Szegedy et al., 2016, Warde-Farley and Goodfellow, 2016] is used.

Based on our theory, we introduce the data-based redaction algorithm (Alg. 1). We then combine this algorithm with an improper active learning algorithm by Hanneke et al. [2018] and introduce the validity-based redaction algorithm (Alg. 2). Finally, we propose to use a guide function to guide the discriminator via a classifier, and introduce the classifier-based redaction algorithm (Alg. 3).

Finally, we empirically evaluate these redaction algorithms via experiments on real-world image datasets (see Section 4). We show that these algorithms can redact quickly while keeping high generation quality. We then investigate applications of data redaction, and use our algorithms to remove different biases that may not exist in the training set but are learned by the pre-trained model. This demonstrates that data redaction can be used to reduce biases and improve generation quality, and hence improve the trustworthiness of generative models.

In summary, our contributions are as follows:

- We formalize the problem of post-editing generative models to prevent them from outputting undesirable samples as "data redaction" and establish its differences with data deletion.
- We propose three data augmentation-based algorithms for redacting data from pre-trained GANs as a function of how the inputs to be redacted are described.
- We theoretically prove that data redaction can be achieved by the proposed algorithms.
- We extensively evaluate our algorithms on real world image datasets. We show these algorithms can redact data quickly while retaining high generation quality. Moreover, we find data redaction performs better than data deletion in a de-biasing experiment.

## 2   A Formal Framework for Data Redaction

Let $p_{\text{data}}$ be the data distribution on $\mathbb{R}^d$ and $X \sim p_{\text{data}}$ be i.i.d. samples. Let $\mathcal{A}$ be the learning algorithm of generative modelling and $\mathcal{M} = \mathcal{A}(X)$ be the pre-trained model on $X$, which learns

$p_{\text{data}}$. In this paper, we consider $\mathcal{A}$ to be a GAN learning algorithm [Goodfellow et al., 2014a], and $\mathcal{M}$ contains two networks, $D$ (discriminator) and $G$ (generator), which are jointly trained to optimize

$$\min_G \max_D \ \mathbb{E}_{x \sim p_{\text{data}}} \log D(x) + \mathbb{E}_{x \sim p_G} \log(1 - D(x)), \tag{1}$$

where $p_G = G \# \mathcal{N}(0, I)$ is defined as the distribution of $G(Z)$ where $Z \sim \mathcal{N}(0, I)$.

## 2.1 Data Redaction Framework

Let the redaction set $\Omega \subset \mathbb{R}^d$ be the set of samples we would like the model to redact. Formally, the goal is to develop a redaction algorithm $\mathcal{D}$ such that $\mathcal{M}' = \mathcal{D}(\mathcal{M}, \Omega)$ learns the data distribution restricted to the complement $\bar{\Omega} = \mathbb{R}^d \setminus \Omega$, i.e. $p_{\text{data}}|_{\bar{\Omega}}$. Examples of $\Omega$ include inconsistent, blurred, unrealistic, or banned samples that are possibly generated by the model.

The redaction set $\Omega$, in addition to the pre-trained model, is considered as an input to the redaction algorithm. We consider three kinds of $\Omega$, namely data-based, validity-based, and classifier-based.

## 2.2 Redaction Set Descriptions

We propose three different descriptions for the redaction set $\Omega$. First, the data-based $\Omega$ is a pre-defined set of samples in $\mathbb{R}^d$, such as a transformation applied on all training samples [Sinha et al., 2020]. Second, the validity-based $\Omega$ is defined as all invalid samples according to a validity function $\mathbf{v} : \mathbb{R}^d \to \{0, 1\}$, where $0$ means invalid and $1$ means valid. This is similar to the setting in Hanneke et al. [2018]. Finally, let $\mathbf{f} : \mathbb{R}^d \to [0, 1]$ be a soft classifier that outputs the probability that a sample belongs to a certain binary class, and $\tau \in (0, 1)$ be a threshold. Then, the classifier-based $\Omega$ is defined as $\{x : \mathbf{f}(x) < \tau\}$. For example, $\mathbf{f}$ can be an offensive text classifier in language generation tasks [Pitsilis et al., 2018].

## 2.3 Data Deletion versus Data Redaction

Motivated by privacy laws such as the GDPR and the CCPA, there has been a recent body of work on data deletion or machine unlearning [Cao and Yang, 2015, Guo et al., 2019, Schelter, 2020, Neel et al., 2021, Sekhari et al., 2021, Izzo et al., 2021, Ullah et al., 2021]. In data deletion, we are given a subset set $X' \subset X$ of the training set to be deleted from an already-trained model, and the goal is to approximate the re-trained model $\mathcal{A}(X \setminus X')$. While there are some superficial similarities – in that the goal is to post-edit models in order to "remove" a few data points, there are two key differences.

The first is that data redaction requires the model to assign zero likelihood to the redaction set $\Omega$ in order to avoid generating samples from this region; this is not the case in data deletion – in fact, we present an example below which shows that data deletion of a set $X'$ may not cause a generative model to redact $X'$.

Specifically, in Fig. 1, the entire data distribution $p_{\text{data}} = \mathcal{N}(0, 1)$ (blue line) is the standard Gaussian distribution on $\mathbb{R}$. We set the redaction set $\Omega = (-\infty, -1.5] \cup [1.5, \infty)$, so the blue samples fall in $\Omega$ and orange samples outside. The learning algorithm $\mathcal{A}$ is the maximum likelihood Gaussian learner that fits the mean and variance of the data. With $n = 80$ samples, the learnt density $\mathcal{A}(X)$ is shown in green. If the blue samples were **deleted**, and the model re-fitted, the newly learnt density $\mathcal{A}(X \setminus X')$ would be the red line. Notice that this red line has considerable density on the blue points – and so these points are not redacted. In contrast, the correct **redaction** solution that redacts the samples in $\Omega$ would be the orange density. Thus deletion does not necessarily lead to redaction.

The second difference is that the redaction set $\Omega$ may have a *zero intersection* with the training data, but may appear in the generated data due to artifacts of the model. Examples include unnatural sounds emitted by speech models, and blurred images from image models. Data redaction, in contrast to data deletion, can address this challenge.

## 3 Methods

In this section, we describe algorithms for each kind of redaction set described in Section 2. We also provide theory on the optimality of the generator and the discriminator. Finally, we generalize the algorithms to situations where we would like the model to redact the union of multiple redaction sets.

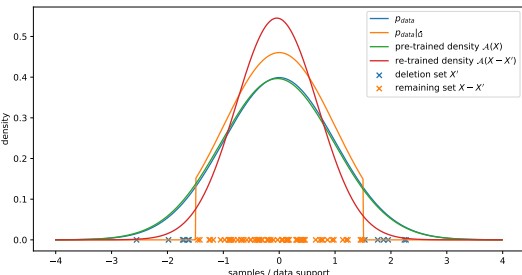

Figure 1: An example showing difference between data redaction and data deletion. The goal of **data deletion** is to approximate the re-trained model (red density), while the goal of **data redaction** is to approximate the restricted density (orange density).

### 3.1 Data-based Redaction Set

The data-based redaction set $\Omega$ is a pre-defined set of samples we would like the model to redact. One example is a transformation function `NegAug` applied to all training samples, where `NegAug` makes realistic images unrealistic or inconsistent [Sinha et al., 2020]. Another example can be visually nice samples outside data manifold when the training set is small [Asokan and Seelamantula, 2020].

In our framework, the redaction set $\Omega$ can be any set of carefully designed or selected samples depending on the purpose of redacting them – which includes but does not limit to improving the generation quality of the model. For example, we expect the model to improve on fairness, bias, ethics or privacy when $\Omega$ is properly constructed with unfair, biased, unethical, or atypical samples.

To redact $\Omega$, we regard both generated samples and $\Omega$ to be fake samples, and all training samples that are not in $\Omega$ to be real samples [Sinha et al., 2020, Asokan and Seelamantula, 2020]. Let $p_\Omega$ be a distribution such that $\mathbf{supp}(p_\Omega) = \Omega$. Then, the fake data distribution $p_{\text{fake}}$ is a mixture of the generative distribution $p_G$ and the redaction distribution $p_\Omega$:

$$p_{\text{fake}} = \lambda \cdot p_G + (1 - \lambda) \cdot p_\Omega, \tag{2}$$

where $\lambda \in (0, 1)$ is a hyperparameter. We also apply label smoothing [Salimans et al., 2016, Szegedy et al., 2016, Warde-Farley and Goodfellow, 2016] techniques to the minimax loss function in order to improve robustness of the discriminator. Let $\alpha_+ \in (\frac{1}{2}, 1]$ be the positive target (such as $0.9$) and $\alpha_- \in [0, \frac{1}{2})$ be the negative target (such as $0.1$). Then, the loss function is

$$
\begin{aligned}
L(G, D) = \quad & \mathbb{E}_{x \sim p_{\text{data}}|_{\bar{\Omega}}} \left[ \alpha_+ \log D(x) + (1 - \alpha_+) \log(1 - D(x)) \right] \\
+ \quad & \mathbb{E}_{x \sim p_{\text{fake}}} \left[ \alpha_- \log D(x) + (1 - \alpha_-) \log(1 - D(x)) \right].
\end{aligned} \tag{3}
$$

**Theorem 1.** *The optimal solution to $\min_G \max_D L(G, D)$ is*

$$D^* = \frac{\alpha_+ p_{\text{data}}|_{\bar{\Omega}} + \alpha_-(\lambda p_G + (1 - \lambda) p_\Omega)}{p_{\text{data}}|_{\bar{\Omega}} + \lambda p_G + (1 - \lambda) p_\Omega}, \quad p_{G^*} = p_{\text{data}}|_{\bar{\Omega}} . \tag{4}$$

We provide the proof and theoretical extension to the more general $f$-GAN [Nowozin et al., 2016] setting in Appendix B. In the data-based setting, we let $p_\Omega = \mathcal{U}(\Omega)$, the uniform distribution on $\Omega$. We assume $\Omega$ has positive, finite Lebesgue measure in $\mathbb{R}^d$ so that $\mathcal{U}(\Omega)$ is well-defined. The proposed method is summarized in Alg. 1.

Our objective function is connected to Sinha et al. [2020] and Asokan and Seelamantula [2020] in the sense that $p_\Omega$ is an instance of the negative distribution described in their frameworks. However, there are several significant differences between our method and theirs: (1) we start from a pre-trained model, (2) we aim to learn $p_{\text{data}}|_{\bar{\Omega}}$ rather than $p_{\text{data}}$ and therefore do not require $\Omega \cap \mathbf{supp}(p_{\text{data}})$ to be the empty set, and (3) we use the common label smoothing techniques and provide theory for this setting. These differences are also true in the following sections.

### 3.2 Validity-based Redaction Set

Let $\mathbf{v} : \mathbb{R}^d \to \{0, 1\}$ be a validity function that indicates whether a sample is valid. Then, validity-based redaction set $\Omega$ is the set of all invalid samples $\{x : \mathbf{v}(x) = 0\}$. For example, $\mathcal{M}$ is a code

---

**Algorithm 1** Redaction Algorithm for Data-based Redaction Set

---

**Inputs**: Pre-trained model $\mathcal{M} = (G_0, D_0)$, train set $X$, redaction set $\Omega$.
Initialize $G = G_0$, $D = D_0$.
Define the fake data distribution $p_{\text{fake}}$ according to (2) with $p_\Omega = \mathcal{U}(\Omega)$.
Train $G, D$ to optimize (3): $\min_G \max_D L(G, D)$.
**return** $\mathcal{M}' = (G, D)$.

---

---

**Algorithm 2** Redaction Algorithm for Validity-based Redaction Set

---

**Inputs**: Pre-trained model $\mathcal{M} = (G_0, D_0)$, train set $X$, validity function $\mathbf{v}$.
Initialize $\Omega' = \{x \in X : \mathbf{v}(x) = 0\}$, $\mathcal{M}_0 = \mathcal{M}$.
**for** $i = 0, \cdots, R - 1$ **do**
  Initialize $G = G_i$, $D = D_i$. Draw $T$ samples $X_{\text{gen}}^{(i)}$ from $G_i$.
  Query $\mathbf{v}$ and add invalid samples to $\Omega'$: $\Omega' \leftarrow \Omega' \cup \{x \in X_{\text{gen}}^{(i)} : \mathbf{v}(x) = 0\}$.
  Define the fake data distribution $p_{\text{fake}}$ according to (2) with $p_\Omega = \mathcal{U}(\Omega')$.
  Let $\mathcal{M}_{i+1} = (G_{i+1}, D_{i+1})$ optimize (3): $\min_G \max_D L(G, D)$.
**end for**
**return** $\mathcal{M}' = (G_R, D_R)$

---

---

**Algorithm 3** Redaction Algorithm for Classifier-based Redaction Set

---

**Inputs**: Pre-trained model $\mathcal{M} = (G_0, D_0)$, train set $X$, differentiable classifier $\mathbf{f}$.
Initialize $G = G_0$, $D = D_0$.
Define the fake data distribution $p_{\text{fake}}$ according to (2) with $p_\Omega = \mathcal{U}(\{x \in X : \mathbf{f}(x) < \tau\})$.
Train $G, D$ to optimize (3): $\min_G \max_D L(G, \texttt{guide}(D, \mathbf{f}))$, where $\texttt{guide}(\cdot, \cdot)$ is defined in (6).
**return** $\mathcal{M}' = (G, D)$.

---

generation model, and $\mathbf{v}$ is a compiler that indicates whether the code is free of syntax errors [Hanneke et al., 2018]. Different from the data-based setting, the validity-based $\Omega$ may have infinite Lebesgue measure, such as a halfspace, and consequently $\mathcal{U}(\Omega)$ may not be well-defined.

To redact $\Omega$, we let $p_\Omega$ in (2) to be a mixture of $p_{\text{data}}|_\Omega$ and $p_G|_\Omega$. This corresponds to a simplified version of the improper active learning algorithm introduced by Hanneke et al. [2018] with our Alg. 1 as their optimization oracle. The idea is to apply Alg. 1 for $R$ rounds. After each round, we query the validity of $T$ newly generated samples and use invalid samples to form a data-based redaction set $\Omega'$. In contrast to the data-based approach, this active algorithm focuses on invalid samples that are more likely to be generated, and therefore efficiently penalizes generation of invalid samples. The proposed method is summarized in Alg. 2.

The total number of queries to the validity function $\mathbf{v}$ is $|X| + T \times R$. In case $\mathbf{v}$ is expensive to run, we would like to achieve better data redaction within a limited number of queries. From the data-driven point of view, we hope to collect as many invalid samples as possible. This is done by setting $R = 1$ and $T$ maximized if we assume less invalid samples are generated after each iteration. However, this may not be the case in practice. We hypothesis some samples are easier to redact while others harder. By setting $R > 1$, we expect an increasing fraction of invalid generated samples to be hard to redact after each iteration. Focusing on these hard samples can potentially help the generator redact them. Since it is hard to directly analyze neural networks, we leave the rigorous study to future work. In Appendix C, we study a much simplified dynamical system corresponding to Alg. 2, where we show the invalidity (the mass of $p_G$ on $\Omega$) converges to zero, and provide optimal $T$ and $R$ values.

### 3.3 Classifier-based Redaction Set

We would like the model to redact samples with certain (potentially undesirable) property. Let $\mathbf{f} : \mathbb{R}^d \to [0, 1]$ be a soft binary classifier on the property (0 means having the property and 1 means not having it), and $\tau \in (0, 1)$ be a threshold. The classifier-based redaction set $\Omega$ is then defined as $\{x : \mathbf{f}(x) < \tau\}$. For example, the property can be *being offensive* in language generation, *containing no speech* in speech synthesis, or *visual inconsistency* in image generation. We consider $\mathbf{f}$ to be a trained machine learning model that is fully accessible and differentiable.

To redact $\Omega$, we let $p_\Omega$ be a mixture of $p_{\text{data}}|_\Omega$ and $p_G|_\Omega$, similar to the validity-based approach. We use $\mathbf{f}$ to guide the discriminator and make it able to easily detect samples from $\Omega$. Let $\texttt{guide}(D, \mathbf{f})$ be a guided discriminator that assigns small values to $x$ when $\mathbf{f}(x) < \tau$ or $D(x)$ is small (i.e. $x \sim p_{\text{fake}}$), and large values to $x$ when $\mathbf{f}(x) > \tau$ and $D(x)$ is large (i.e. $x \sim p_{\text{data}}|_{\bar{\Omega}}$). Instead of optimizing $L(G, D)$ in (3), we optimize $L(G, \texttt{guide}(D, \mathbf{f}))$. This will effectively update $G$ by preventing it from generating samples in $\Omega$. According to **Theorem** 1, the optimal discriminator is the solution to

$$\texttt{guide}(D^*, \mathbf{f}) = \frac{\alpha_+ p_{\text{data}}|_{\bar{\Omega}} + \alpha_-(\lambda p_G + (1 - \lambda)p_\Omega)}{p_{\text{data}}|_{\bar{\Omega}} + \lambda p_G + (1 - \lambda)p_\Omega}. \tag{5}$$

Therefore, the design of the $\texttt{guide}$ function must make (5) feasible. In this paper, we let

$$\texttt{guide}(D, \mathbf{f})(x) = \begin{cases} D(x) & \text{if } \mathbf{f}(x) \geq \tau \\ \alpha_- + (D(x) - \alpha_-)\mathbf{f}(x) & \text{otherwise} \end{cases}. \tag{6}$$

The feasibility of (5) is discussed in Appendix D. The proposed method is summarized in Alg. 3. The classifier-based $\Omega$ generalizes the validity-based $\Omega$. First, any validity-based $\Omega$ can be represented by a classifier-based $\Omega$ if we let $\mathbf{f} = \mathbf{v}$ and $\tau = \frac{1}{2}$. Next, we note there is a trivial way to deal with classifier-based $\Omega$ via the validity-based approach – by setting $\mathbf{v}(x) = 1\{\mathbf{f}(x) < \tau\}$. However, potentially useful information such as values and gradients of $\mathbf{f}$ are lost, and we will evaluate this effect in experiments. In addition, the classifier-based approach does not maintain the potentially large set of invalid generated samples, as this step is automatically done in the $\texttt{guide}$ function.

### 3.4 Generalization to Multiple Redaction Sets

Let $\{\Omega_k\}_{k=1}^K$ be disjoint sets in $\mathbb{R}^d$, and we would like the model to redact $\Omega = \bigcup_{k=1}^K \Omega_k$. In the data-based setting, we let $p_\Omega = \mathcal{U}(\Omega) = \mathcal{U}(\bigcup_{k=1}^K \Omega_k)$. In the validity-based setting, each $\Omega_k$ is associated with a validity function $\mathbf{v}_k$. We let the overall validity function to be $\mathbf{v}(x) = \min_k \mathbf{v}_k(x)$. In the classifier-based setting, each $\Omega_k$ is associated with a classifier $\mathbf{f}_k$. Similar to the validity-based setting, we let the overall $\mathbf{f}$ to be $\mathbf{f}(x) = \min_k \mathbf{f}_k(x)$.

## 4 Experiments

In this section, we aim to answer the following questions.

- How well can the algorithms in Section 3 redact samples in practice?
- Can these algorithms be used to de-bias pre-trained models?
- Can these algorithms be used to understand training data?

We examine these questions by focusing on several real-world image datasets, including MNIST ($28 \times 28$) [LeCun et al., 2010], CIFAR ($32 \times 32$) [Krizhevsky et al., 2009], CelebA ($64 \times 64$) [Liu et al., 2015] and STL-10 ($96 \times 96$) [Coates et al., 2011] datasets. We demonstrate main experiments in Appendix E, and provide more detailed results afterwards. Specifically, in Appendix E.2 and F, we investigate how well these algorithms can redact samples with a specific label. In Appendix E.3 and G, we investigate how well these algorithms can de-bias pre-trained models and improve generation quality. In Appendix E.4 and H, we use these algorithms to understand training data through the lens of data redaction.

## 5 Conclusion

In this paper, we propose a systematic framework for redacting data from pre-trained generative models. We provide three different algorithms for GANs that differ on how the samples to be redacted are described. We provide theoretical results that data redaction can be achieved. We then empirically investigate data redaction on real-world image datasets, and show that our algorithms are capable of redacting data while retaining high generation quality at a fraction of the cost of full re-training. One limitation or our paper is that the proposed framework only applies to unconditional generative models. It is an important future direction to define data redaction and propose algorithms for conditional generative models, which are more widely used in downstream deep learning applications.

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

# A   Related Work

Although deep generative models have been highly successful at many domains, it has long been known that they often emit undesirable samples and samples with different types of artifacts that make them untrustworthy. Examples include blurred image samples [Kaneko and Harada, 2021], fairness issues [Tan et al., 2020, Karakas et al., 2022], and checkerboard artifacts [Odena et al., 2016, Zhang et al., 2019, Wang et al., 2020, Schwarz et al., 2021] in image generation, offensive text in language models [Abid et al., 2021, Perez et al., 2022], and unnatural sound in speech models [Donahue et al., 2018, Thiem et al., 2020].

Some prior works have used post-editing to remove artifacts and improve GANs. Examples include improving fairness [Tan et al., 2020, Karakas et al., 2022], rule rewriting [Bau et al., 2020], discovering interpretability [Härkönen et al., 2020], and fine-tuning [Mo et al., 2020, Li et al., 2020, Zhao et al., 2020]. The purpose, use cases, and editing methods of these papers are different from our paper, where we focus on data redaction.

While our problem definition and formalization is novel, the technical solutions that we propose are related to three prior works that use these techniques in different contexts. These are NDA [Sinha et al., 2020], Rumi-GAN [Asokan and Seelamantula, 2020], and Hanneke et al. [2018]. The first two papers look at how to avoid generating negative samples while training a generative model from scratch. This is done by defining new fake distributions to penalize the generation of these samples. However, their purposes are different from us: NDA is used to characterize the boundary of the support of the generative distribution more precisely, and Rumi-GAN is used to handle unbalanced data. We extend their idea and theory to data redaction in Section 3. [1] Hanneke et al. [2018] propose an active learning approach to avoid generating invalid samples, also while training a generative model from scratch. Their work however is entirely theoretical and apply to discrete distributions. In our paper, the validity-based redaction algorithm (Alg. 2) is based on a simplified version of their algorithm. We also use their definition of *invalidity* as an evaluation method.

Our work is also related to data deletion or machine unlearning [Cao and Yang, 2015, Guo et al., 2019, Schelter, 2020, Neel et al., 2021, Sekhari et al., 2021, Izzo et al., 2021, Ullah et al., 2021]. However, there are two important differences between data deletion and data redaction. First, data deletion aims to approximate the re-trained model when some training samples are removed – mostly due to privacy reasons – while in data redaction we penalize the model from knowing samples that should be redacted. Another difference is that in data redaction, the redaction set $\Omega$ may have a zero intersection with training data. These two differences are discussed in Section 2.3 in detail. In addition, most data deletion techniques are for supervised learning or clustering, and is much less studied for generative models.

There is also a related line of work on catastrophic forgetting in supervised learning [Kirkpatrick et al., 2017] and generative models [Thanh-Tung and Tran, 2020]. This concept is different from data redaction in that we would like the generative model to redact certain data after training, while catastrophic forgetting means knowledge learned in previous tasks is destroyed during continual learning.

---

[1] The loss functions in NDA and Rumi-GAN are similar.

# B Proof of Theorem 1 and Extension to $f$-GAN

**Background of $f$-GAN [Nowozin et al., 2016].** Let $\phi$ be a convex, lower-semicontinuous function such that $\phi(1) = 0$. In $f$-GAN, the following $\phi$-divergence is minimized:

$$D_\phi(P\|Q) = \int_{x\in\mathbb{R}^d} Q(x)\phi\left(\frac{P(x)}{Q(x)}\right)dx.$$

According to the variational characterization of $\phi$-divergence [Nguyen et al., 2010],

$$D_\phi(P\|Q) = \sup_T \left[\mathbb{E}_{x\sim P}T(x) - \mathbb{E}_{x\sim Q}\phi^*(T(x))\right],$$

where the optimal $T$ is obtained by $T = \phi'\left(\frac{P}{Q}\right)$.

**The objective function (3) corresponds to an $f$-GAN.** Let $\alpha = \alpha_- + \alpha_+$. We can rewrite (3) as

$$L(G, D) = \alpha \cdot \mathbb{E}_{x\sim P}\log D(x) + (2 - \alpha) \cdot \mathbb{E}_{x\sim Q}\log(1 - D(x)),$$

where

$$P = \frac{\alpha_+}{\alpha}p_{\text{data}}|_{\bar\Omega} + \frac{\alpha_-}{\alpha}p_{\text{fake}}; \quad Q = \frac{1-\alpha_+}{2-\alpha}p_{\text{data}}|_{\bar\Omega} + \frac{1-\alpha_-}{2-\alpha}p_{\text{fake}}.$$

Let

$$C = \alpha\log\alpha + (2-\alpha)\log(2-\alpha) - 2\log 2,$$

$$\phi(u) = (\alpha u)\log(\alpha u) - (\alpha u - \alpha + 2)\log(\alpha u - \alpha + 2) + (2 - \alpha)\log(2 - \alpha) - C.$$

Then, $\phi(1) = 0$, and $\phi''(u) = \frac{\alpha(2-\alpha)}{u(\alpha u - \alpha + 2)} > 0$ so $\phi$ is convex. Its convex conjugate function $\phi^*$ is

$$\phi^*(t) := \sup_u(ut - \phi(u)) = -(2 - \alpha)\log\left(1 - e^{\frac{t}{\alpha}}\right) + C.$$

Let $T(x) = \alpha\log D(x)$. Then,

$$\max_D L(G, D) = \sup_T \left[\mathbb{E}_{x\sim P}T(x) - \mathbb{E}_{x\sim Q}\phi^*(T(x))\right] + C = D_\phi(P\|Q) + C.$$

**Optimal $D$.** We have

$$\phi'(u) = \alpha\log\frac{\alpha u}{\alpha u - \alpha + 2}.$$

Therefore, the optimal discriminator is

$$\alpha\log D = \phi'\left(\frac{P}{Q}\right),$$

or

$$D = \frac{\alpha P}{\alpha P + (2 - \alpha)Q} = \frac{\alpha_+ p_{\text{data}}|_{\bar\Omega} + \alpha_- p_{\text{fake}}}{p_{\text{data}}|_{\bar\Omega} + p_{\text{fake}}}.$$

Finally, the optimal discriminator in (4) is obtained by inserting (2) into the above equation.

**Optimal $G$.** For conciseness, we let

$$P_1 = p_{\text{data}}|_{\bar\Omega}, P_2 = p_G, P_3 = p_\Omega,$$

$$\beta_1 = \frac{\alpha_+}{\alpha}, \beta_2 = \frac{\alpha_-\lambda}{\alpha}, \beta_3 = \frac{\alpha_-(1-\lambda)}{\alpha},$$

$$\gamma_1 = \frac{1-\alpha_+}{2-\alpha}, \gamma_2 = \frac{(1-\alpha_-)\lambda}{2-\alpha}, \gamma_3 = \frac{(1-\alpha_-)(1-\lambda)}{2-\alpha}.$$

Then, we have

$$P = \sum_{i=1}^3 \beta_i P_i, Q = \sum_{i=1}^3 \gamma_i P_i.$$

We also have

$$\frac{\beta_1}{\gamma_1} > \frac{\beta_2}{\gamma_2} = \frac{\beta_3}{\gamma_3}.$$

Because $\mathbf{supp}(P_1) \cap \mathbf{supp}(P_3)$ is the empty set, we have

$$D_\phi(P\|Q) = \int_{x\in\mathbb{R}^d} \left(\sum_{i=1}^3 \gamma_i P_i\right) \phi\left(\frac{\sum_{i=1}^3 \beta_i P_i}{\sum_{i=1}^3 \gamma_i P_i}\right) dx$$

$$= \int_{x\notin\Omega} (\gamma_1 P_1 + \gamma_2 P_2)\phi\left(\frac{\beta_1 P_1 + \beta_2 P_2}{\gamma_1 P_1 + \gamma_2 P_2}\right) dx$$

$$+ \int_{x\in\Omega} (\gamma_2 P_2 + \gamma_3 P_3)\phi\left(\frac{\beta_2 P_2 + \beta_3 P_3}{\gamma_2 P_2 + \gamma_3 P_3}\right) dx$$

Let

$$\int_{x\in\Omega} P_2 dx = \eta.$$

We have

$$\int_{x\in\Omega} (\gamma_2 P_2 + \gamma_3 P_3)\phi\left(\frac{\beta_2 P_2 + \beta_3 P_3}{\gamma_2 P_2 + \gamma_3 P_3}\right) dx = (\gamma_2\eta + \gamma_3)\,\phi\left(\frac{\beta_3}{\gamma_3}\right).$$

Let

$$\zeta = \frac{\beta_2(\gamma_1 + \gamma_2)}{\gamma_2(\beta_1 + \beta_2)}.$$

According to Jensen's inequality,

$$\int_{x\notin\Omega} (\gamma_1 P_1 + \gamma_2 P_2)\phi\left(\frac{\beta_1 P_1 + \beta_2 P_2}{\gamma_1 P_1 + \gamma_2 P_2}\right) dx$$

$$= (\gamma_1 + \gamma_2(1 - \zeta\eta))\int_{x\notin\Omega} \left(\frac{\gamma_1 P_1 + \gamma_2 P_2}{\gamma_1 + \gamma_2(1-\zeta\eta)}\right) \phi\left(\frac{\beta_1 P_1 + \beta_2 P_2}{\gamma_1 P_1 + \gamma_2 P_2}\right) dx$$

$$\geq (\gamma_1 + \gamma_2(1 - \zeta\eta))\phi\left(\int_{x\notin\Omega} \frac{\beta_1 P_1 + \beta_2 P_2}{\gamma_1 + \gamma_2(1-\zeta\eta)} dx\right)$$

$$= (\gamma_1 + \gamma_2(1 - \zeta\eta))\phi\left(\frac{\beta_1 + \beta_2(1-\eta)}{\gamma_1 + \gamma_2(1-\zeta\eta)}\right)$$

$$= (\gamma_1 + \gamma_2(1 - \zeta\eta))\phi\left(\frac{\beta_1 + \beta_2}{\gamma_1 + \gamma_2}\right).$$

Therefore, we have

$$D_\phi(P\|Q) \geq (\gamma_1+\gamma_2)\phi\left(\frac{\beta_1 + \beta_2}{\gamma_1 + \gamma_2}\right) + \gamma_3\phi\left(\frac{\beta_3}{\gamma_3}\right) + \left[\gamma_2\phi\left(\frac{\beta_3}{\gamma_3}\right) - \frac{\beta_2(\gamma_1 + \gamma_2)}{\beta_1 + \beta_2}\phi\left(\frac{\beta_1 + \beta_2}{\gamma_1 + \gamma_2}\right)\right]\eta.$$

Now, we show the $\eta$ term is non-negative. We write

$$\gamma_2\phi\left(\frac{\beta_3}{\gamma_3}\right) - \frac{\beta_2(\gamma_1 + \gamma_2)}{\beta_1 + \beta_2}\phi\left(\frac{\beta_1 + \beta_2}{\gamma_1 + \gamma_2}\right) = \beta_2\left(\frac{\gamma_2}{\beta_2}\phi\left(\frac{\beta_3}{\gamma_3}\right) - \frac{(\gamma_1 + \gamma_2)}{\beta_1 + \beta_2}\phi\left(\frac{\beta_1 + \beta_2}{\gamma_1 + \gamma_2}\right)\right)$$

$$= \beta_2\left(\frac{\gamma_3}{\beta_3}\phi\left(\frac{\beta_3}{\gamma_3}\right) - \frac{1 - \gamma_3}{1 - \beta_3}\phi\left(\frac{1 - \beta_3}{1 - \gamma_3}\right)\right).$$

It suffices to prove the function $\psi(u) = \phi(u)/u$ satisfies

$$\psi\left(\frac{\beta_3}{\gamma_3}\right) \geq \psi\left(\frac{1 - \beta_3}{1 - \gamma_3}\right).$$

We use the Mathematica software [Inc.] to compute the difference:

$$\psi\left(\frac{\beta_3}{\gamma_3}\right) - \psi\left(\frac{1 - \beta_3}{1 - \gamma_3}\right) = -\frac{\alpha}{\alpha_-}\log\frac{2 - \alpha}{1 - \alpha_-} + \alpha\alpha_-\log\frac{\alpha_-(2 - \alpha)}{1 - \alpha_-} + \frac{\alpha(1 - \alpha_-)}{2 - \alpha}(\log 4 - \alpha\log\alpha)$$

$$-\frac{\alpha}{2 - \alpha}\left(\frac{\lambda + 1}{\lambda\alpha_- + \alpha_+} - 1\right)(\log 4 - \alpha\log\alpha)$$

$$-\alpha\log\frac{(2 - \alpha)(\lambda\alpha_- + \alpha_+)}{\lambda(1 - \alpha_-) + 1 - \alpha_+} + \frac{\alpha(\lambda + 1)}{\lambda\alpha_- + \alpha_+}\log\frac{(\lambda + 1)(2 - \alpha)}{\lambda(1 - \alpha_-) + 1 - \alpha_+}.$$

The minimum value of the above difference for $\alpha_- \in [0, \frac{1}{2}]$, $\alpha_+ \in [0, \frac{1}{2}]$, and $\lambda \in [0, 1]$ is obtained at $\alpha_- = \alpha_+ = \frac{1}{2}$, where the difference equals zero. This makes us able to conclude

$$D_\phi(P\|Q) \geq (\gamma_1 + \gamma_2)\phi\left(\frac{\beta_1 + \beta_2}{\gamma_1 + \gamma_2}\right) + \gamma_3\phi\left(\frac{\beta_3}{\gamma_3}\right).$$

Finally, we let $P_2 = P_1$. In this case,

$$\begin{aligned}
D_\phi(P\|Q) &= \int_{x \in \mathbb{R}^d} \left(\sum_{i=1}^{3} \gamma_i P_i\right) \phi\left(\frac{\sum_{i=1}^{3} \beta_i P_i}{\sum_{i=1}^{3} \gamma_i P_i}\right) dx \\
&= \int_{x \notin \Omega} (\gamma_1 P_1 + \gamma_2 P_2)\phi\left(\frac{\beta_1 P_1 + \beta_2 P_2}{\gamma_1 P_1 + \gamma_2 P_2}\right) dx \\
&\quad + \int_{x \in \Omega} \gamma_3 P_3 \phi\left(\frac{\beta_3 P_3}{\gamma_3 P_3}\right) dx \\
&= (\gamma_1 + \gamma_2)\phi\left(\frac{\beta_1 + \beta_2}{\gamma_1 + \gamma_2}\right) + \gamma_3\phi\left(\frac{\beta_3}{\gamma_3}\right).
\end{aligned}$$

Therefore, the optimal generator is $p_G = p_{\text{data}}|_{\bar{\Omega}}$.

**Extension to $f$-GAN.**  We can extend the objective (3) to any type of $f$-GAN. Let $\phi$ be a convex, lower-semicontinuous function such that $\phi(1) = 0$. Let

$$P = \frac{\alpha_+}{\alpha} p_{\text{data}}|_{\bar{\Omega}} + \frac{\alpha_-}{\alpha} p_{\text{fake}}; \quad Q = \frac{1 - \alpha_+}{2 - \alpha} p_{\text{data}}|_{\bar{\Omega}} + \frac{1 - \alpha_-}{2 - \alpha} p_{\text{fake}}.$$

We jointly optimize

$$\min_G \max_D \; L(G, D) = \mathbb{E}_{x \sim P} D(x) - \mathbb{E}_{x \sim Q} \phi^*(D(x)).$$

Then, the optimal discriminator is $D = \phi'\left(\frac{P}{Q}\right)$. If $\psi\left(\frac{\beta_3}{\gamma_3}\right) \geq \psi\left(\frac{1-\beta_3}{1-\gamma_3}\right)$, then the optimal generator is $p_G = p_{\text{data}}|_{\bar{\Omega}}$.

**Remark 1.** *When $\alpha_- = 0$ and $\alpha_+ = 1$ (i.e. there is no label smoothing), Theorem 1 in Sinha et al. [2020] implies the above optimal generator. Our theorem also extends their theorem to the label smoothing setting.*

## C Theoretical Analysis of a Simplified Dynamical System on Invalidity

In this section, we provide theoretical analysis to a simplified, ideal dynamical system that corresponds to Alg. 2 and Section 3.2. In this dynamical system, we assume there are only two types of invalid samples: those easy to redact, and those hard to redact. We assume after each iteration, the generator will generate a less but positive fraction of invalid samples. Formally, let $\{\Omega_{\text{easy}}, \Omega_{\text{hard}}\}$ be a split of $\Omega$, where $\Omega_{\text{easy}}$ is the set of invalid samples that are easy to redact, and $\Omega_{\text{hard}}$ is the set of invalid samples that are hard to redact. We let

$$m_{\text{easy}} = \int_{\Omega_{\text{easy}}} p_G(x)dx,$$

$$m_{\text{hard}} = \int_{\Omega_{\text{hard}}} p_G(x)dx,$$

$$m_{\text{ratio}} = \frac{m_{\text{easy}}}{m_{\text{easy}} + m_{\text{hard}}}.$$

Then, $m_{\text{easy}}$ is the fraction of invalid generated samples that are easy to redact, and $m_{\text{hard}}$ is the fraction of invalid generated samples that are hard to redact. $m_{\text{easy}} + m_{\text{hard}}$ is the fraction of invalid generated samples over all generated ones, which we call **invalidity**. We use superscript to represent each iteration. We consider the following dynamical system:

$$m_{\text{easy}}^{i+1} = m_{\text{easy}}^i \cdot \eta_{\text{easy}}(m_{\text{ratio}}^i, T),$$

$$m_{\text{hard}}^{i+1} = m_{\text{hard}}^i \cdot \eta_{\text{hard}}(m_{\text{ratio}}^i, T).$$

In other words, the improvement of $m_{\text{easy}}$ and $m_{\text{hard}}$ (in terms of multiplication factor) is only affected by $m_{\text{ratio}}$ and $T$. We make this assumption because in practice, the number of invalid samples to optimize the loss function is always fixed. As for boundary conditions, we assume $m_{\text{easy}}^0 > m_{\text{hard}}^0$. We assume for $\eta \in \{\eta_{\text{easy}}, \eta_{\text{hard}}\}$, $0 < \eta(m, T) \leq 1$, where equality holds only in these situations:

$$\eta(m, 0) = 1, \ \eta_{\text{easy}}(0, T) = 1, \ \eta_{\text{hard}}(1, T) = 1.$$

We also assume a larger $T$ leads to smaller $\eta$, but this effect degrades as $T$ increases:

$$\frac{\partial}{\partial T}\eta(m, T) < 0, \ \frac{\partial^2}{\partial T^2}\eta(m, T) > 0.$$

To distinguish between samples that are easy or hard to redact, we assume

$$\frac{1}{m} \cdot \frac{\partial}{\partial T}\eta_{\text{easy}}(m, T) < \frac{1}{1-m} \cdot \frac{\partial}{\partial T}\eta_{\text{hard}}(m, T) < 0.$$

We can now draw some conclusions below.

**As $i \to \infty$, invalidity converges to 0.** Because $\eta_{\text{easy}}(T) < 1$ and $\eta_{\text{hard}}(T) < 1$ when $T > 0$, we have $m_{\text{easy}}^{i+1} \leq m_{\text{easy}}^i$ and $m_{\text{hard}}^{i+1} \leq m_{\text{hard}}^i$. According to the monotone convergence theorem, there exists $m_{\text{easy}}^\infty \geq 0$ and $m_{\text{hard}}^\infty \geq 0$ such that

$$\lim_{i \to \infty} m_{\text{easy}}^i = m_{\text{easy}}^\infty, \ \lim_{i \to \infty} m_{\text{hard}}^i = m_{\text{hard}}^\infty.$$

We now prove $m_{\text{easy}}^\infty = m_{\text{hard}}^\infty = 0$. If otherwise, there exists $m_{\text{ratio}}^\infty = \frac{m_{\text{easy}}^\infty}{m_{\text{easy}}^\infty + m_{\text{hard}}^\infty}$ such that $m_{\text{ratio}}^i \to m_{\text{ratio}}^\infty$. We then have

$$m_{\text{easy}}^\infty = m_{\text{easy}}^\infty \cdot \eta_{\text{easy}}(m_{\text{ratio}}^\infty, T),$$

$$m_{\text{hard}}^\infty = m_{\text{hard}}^\infty \cdot \eta_{\text{hard}}(m_{\text{ratio}}^\infty, T).$$

If $m_{\text{easy}}^\infty > 0$, then $m_{\text{ratio}}^\infty > 0$, and $\eta_{\text{easy}}(m_{\text{ratio}}^\infty, T) < 1$, contradiction. Similarly, if $m_{\text{hard}}^\infty > 0$, then $m_{\text{ratio}}^\infty < 1$, and $\eta_{\text{hard}}(m_{\text{ratio}}^\infty, T) < 1$, contradiction. Therefore, we conclude both $m_{\text{easy}}^i$ and $m_{\text{hard}}^i$ converge to 0. This indicates the invalidity converges to zero.

**Simplifying the dynamical system.** To further simplify the problem, we make a strong assumption that $\eta$ is linear in $m$. Then, we must have

$$\eta_{\text{easy}}(m, T) = 1 - \xi_{\text{easy}}(T) \cdot m,$$
$$\eta_{\text{hard}}(m, T) = 1 - \xi_{\text{hard}}(T) \cdot (1 - m),$$

where $\xi \in [0, 1], \xi(0) = 0, \xi' > 0, \xi'' < 0$ for $\xi \in \{\xi_{\text{easy}}, \xi_{\text{hard}}\}$. We also have $\xi'_{\text{easy}} > \xi'_{\text{hard}}$ and therefore $\xi_{\text{easy}} > \xi_{\text{hard}}$.

**Optimal $T$ and $R$ from bounds.** We have

$$m_{\text{easy}}^{i+1} + m_{\text{hard}}^{i+1} = m_{\text{easy}}^i + m_{\text{hard}}^i - \frac{\xi_{\text{easy}}(T)(m_{\text{easy}}^i)^2 + \xi_{\text{hard}}(T)(m_{\text{hard}}^i)^2}{m_{\text{easy}}^i + m_{\text{hard}}^i}.$$

Because $\xi_{\text{easy}}(T) \geq \xi_{\text{hard}}(T)$, we have

$$\frac{\xi_{\text{easy}}(T)\xi_{\text{hard}}(T)}{\xi_{\text{easy}}(T) + \xi_{\text{hard}}(T)}(m_{\text{easy}}^i + m_{\text{hard}}^i) \leq \frac{\xi_{\text{easy}}(T)(m_{\text{easy}}^i)^2 + \xi_{\text{hard}}(T)(m_{\text{hard}}^i)^2}{m_{\text{easy}}^i + m_{\text{hard}}^i} \leq \xi_{\text{easy}}(T)(m_{\text{easy}}^i + m_{\text{hard}}^i).$$

This leads to

$$1 - \xi_{\text{easy}}(T) \leq \frac{m_{\text{easy}}^{i+1} + m_{\text{hard}}^{i+1}}{m_{\text{easy}}^i + m_{\text{hard}}^i} \leq 1 - \frac{\xi_{\text{easy}}(T)\xi_{\text{hard}}(T)}{\xi_{\text{easy}}(T) + \xi_{\text{hard}}(T)},$$

and therefore

$$(1 - \xi_{\text{easy}}(T))^R \leq \frac{m_{\text{easy}}^R + m_{\text{hard}}^R}{m_{\text{easy}}^0 + m_{\text{hard}}^0} \leq \left(1 - \frac{\xi_{\text{easy}}(T)\xi_{\text{hard}}(T)}{\xi_{\text{easy}}(T) + \xi_{\text{hard}}(T)}\right)^R.$$

Assume the number of queries, $T \times R$, is fixed. Then, the optimal $T$ from the lower bound is

$$T_{\text{low}}^* = \arg\min_T \frac{1}{T} \log(1 - \xi_{\text{easy}}(T)).$$

By setting the derivative to be zero, we have $T_{\text{low}}^*$ is the solution to

$$-T\xi'_{\text{easy}}(T) = (1 - \xi_{\text{easy}}(T)) \log(1 - \xi_{\text{easy}}(T)).$$

Similarly, the optimal $T$ from the upper bound is

$$T_{\text{upp}}^* = \arg\min_T \frac{1}{T} \log\left(1 - \frac{\xi_{\text{easy}}(T)\xi_{\text{hard}}(T)}{\xi_{\text{easy}}(T) + \xi_{\text{hard}}(T)}\right).$$

By setting the derivative to be zero, we have $T_{\text{upp}}^*$ is the solution to

$$-T \cdot \frac{\xi'_{\text{easy}}(T)\xi_{\text{hard}}(T)^2 + \xi'_{\text{hard}}(T)\xi_{\text{easy}}(T)^2}{(\xi_{\text{easy}}(T) + \xi_{\text{hard}}(T))^2} = \left(1 - \frac{\xi_{\text{easy}}(T)\xi_{\text{hard}}(T)}{\xi_{\text{easy}}(T) + \xi_{\text{hard}}(T)}\right) \log\left(1 - \frac{\xi_{\text{easy}}(T)\xi_{\text{hard}}(T)}{\xi_{\text{easy}}(T) + \xi_{\text{hard}}(T)}\right).$$

# D   Feasibility of Discriminator in the Classifier-based Setting

The solution to (5) and (6) is:

$$D^*(x) = \begin{cases} \frac{\alpha_+ p_{\text{data}}|_{\bar{\Omega}} + \alpha_-(\lambda p_G + (1-\lambda)p_\Omega)}{p_{\text{data}}|_{\bar{\Omega}} + \lambda p_G + (1-\lambda)p_\Omega} & \text{if } \mathbf{f}(x) \geq \tau \\ \alpha_- & \text{if } \mathbf{f}(x) < \tau \end{cases},$$

which satisfies $D^* \in [0, 1]$. Therefore, (5) is feasible with the `guide` function defined in (6).

# E   Experiments (Main)

In this section, we aim to answer the following questions.

- How well can the algorithms in Section 3 redact samples in practice?
- Can these algorithms be used to de-bias pre-trained models?
- Can these algorithms be used to understand training data?

We examine these questions by focusing on several real-world image datasets, including MNIST ($28 \times 28$) [LeCun et al., 2010], CIFAR ($32 \times 32$) [Krizhevsky et al., 2009], CelebA ($64 \times 64$) [Liu et al., 2015] and STL-10 ($96 \times 96$) [Coates et al., 2011] datasets. We demonstrate main experiments in Appendix 4, and provide more detailed results afterwards. Specifically, in Appendix E.2 and F, we investigate how well these algorithms can redact samples with a specific label. In Appendix E.3 and G, we investigate how well these algorithms can de-bias pre-trained models and improve generation quality. In Appendix E.4 and H, we use these algorithms to understand training data through the lens of data redaction.

The pre-trained model for each dataset is a DCGAN [Radford et al., 2015] trained for 200 epochs (see details in Appendix E.1). We use one NVIDIA 3080 GPU to train these models and run experiments.

**Evaluation Metrics: invalidity and generation quality.** The invalidity is defined as the mass of the generation distribution on the redaction set $\Omega$: $\texttt{Inv}(p_G) = \int_{x \in \Omega} p_G(x)dx$. In practice, we measure invalidity by generating 50K samples and computing the fraction of these samples that fall into $\Omega$.

The generation quality is measured in Inception Score (IS) [Salimans et al., 2016] and Frechet Inception Distance (FID) [Heusel et al., 2017]. Higher IS or lower FID indicates better quality. We compute IS for grey-scale images and FID for RGB images. When measuring quality, we compute IS or FID between 50K generated samples and $X \cap \bar{\Omega}$. Therefore, this score is not comparable with the score w.r.t. the pre-trained model if the redaction set includes samples in the training set, such as samples with a specific label in Appendix E.2. Detailed setup is in Appendix E.1.

## E.1   Experimental Setup

**Pre-training.** We use DCGAN [Radford et al., 2015] with latent dimension $= 128$ as the model. The pre-trained model is trained with label smoothing ($\alpha_+ = 0.9, \alpha_- = 0.1$):

$$\min_G \max_D \quad \mathbb{E}_{x \sim X} \left[ \alpha_+ \log D(x) + (1 - \alpha_+) \log(1 - D(x)) \right]$$
$$+ \mathbb{E}_{z \sim \mathcal{N}(0,I)} \left[ \alpha_- \log D(G(z)) + (1 - \alpha_-) \log(1 - D(G(z))) \right].$$

We use Adam optimizer with learning rate $= 2 \times 10^{-4}, \beta_1 = 0.5, \beta_2 = 0.999$ to optimize both the generator and the discriminator. The networks are trained for 200 epochs with a batch size of 64. For each iteration over one mini-batch, we let $K_D$ be the number of times to update the discriminator, and $K_G$ the number of times to update the generator. We use $K_D = 1$ and $K_G = 5$ to train.

**Data redaction.** The setup is similar to the pre-training except for two differences. The number of epochs is much smaller: 8 for MNIST, 30 for CIFAR, and 40 for STL-10. We let $K_G = 1$ for MNIST and CIFAR and $K_G = 5$ for STL-10.

**Evaluation.** To measure invalidity, we generate 50K samples, and compute the fraction of these samples that are not valid (e.g., classified as the label to be redacted, or with pre-defined biases). It is the lower the better. The invalidity for redacting labels is measured based on label classifiers. We use pre-trained classifiers on these datasets. [2]

The other evaluation metric is generation quality. The inception score (IS) [Salimans et al., 2016] is computed based on logit distributions from the above pre-trained classifiers. It is the higher the better. The Frechet Inception Distance (FID) [Heusel et al., 2017] is computed based on an open-sourced PyTorch implementation. [3] It is the lower the better.

When computing these quality metrics, we generate 50K samples, and compare to the set of valid training samples: $\{x \in X : x \notin \Omega\}$. Therefore, when $X \cap \Omega$ is not the empty set (such as

---

[2] https://github.com/aaron-xichen/pytorch-playground (MIT license)
[3] https://github.com/mseitzer/pytorch-fid (Apache-2.0 license)

redacting labels in Appendix E.2), the quality measure of the model after data redaction is not directly comparable to the pre-trained model, but these scores among different redaction algorithms are comparable and give intuition to the generation quality. When $X \cap \Omega$ is the empty set (such as de-biasing in Appendix E.3), the quality measures of the pre-trained model and the model after data redaction are directly comparable.

## E.2 Redacting Labels

Table 1: Invalidity and generation quality of different redaction algorithms on redacting label zero within different datasets. Mean and standard errors are reported for five random seeds. Note that quality measure after data redaction is not directly comparable with the pre-trained model. The invalidity drops in magnitude after data redaction. Different redaction algorithms are highly comparable to each other.

| Dataset | Evaluation | Pre-trained | Data-based | Validity-based | Classifier-based |
|---|---|---|---|---|---|
| MNIST | $\texttt{Inv}(\downarrow)(\times 10^{-5})$ | $1.1 \times 10^4$ | $8.0 \pm 2.2$ | $6.4 \pm 0.8$ | $\mathbf{5.2 \pm 3.7}$ |
| (8 epochs) | $\texttt{IS}(\uparrow)$ | $7.82$ | $\mathbf{7.20 \pm 0.08}$ | $7.19 \pm 0.04$ | $7.16 \pm 0.04$ |
| CIFAR-10 | $\texttt{Inv}(\downarrow)(\times 10^{-3})$ | $1.3 \times 10^2$ | $\mathbf{7.5 \pm 1.1}$ | $7.6 \pm 1.0$ | $11.6 \pm 1.0$ |
| (30 epochs) | $\texttt{FID}(\downarrow)$ | $36.2$ | $34.8 \pm 1.5$ | $34.8 \pm 1.4$ | $\mathbf{33.2 \pm 0.6}$ |
| STL-10 | $\texttt{Inv}(\downarrow)(\times 10^{-4})$ | $6.2 \times 10^2$ | $8.8 \pm 4.5$ | $\mathbf{7.7 \pm 1.3}$ | $11.6 \pm 3.6$ |
| (40 epochs) | $\texttt{FID}(\downarrow)$ | $79.1$ | $77.8 \pm 2.2$ | $\mathbf{77.0 \pm 2.3}$ | $77.2 \pm 1.5$ |

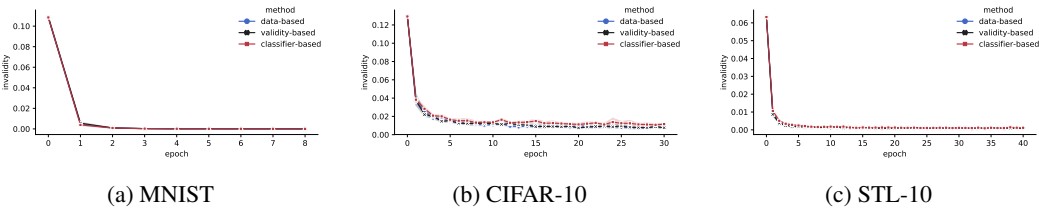

(a) MNIST        (b) CIFAR-10        (c) STL-10

Figure 2: Invalidity during data redaction when redacting label zero. Mean and standard errors are plotted for five random seeds. Standard errors may be too small to spot. Invalidity drops quickly at the beginning of data redaction, and different algorithms are highly comparable to each other.

**Question.** How well can the algorithms in Section 3 redact samples in practice?

**Methodology.** We investigate how well the proposed algorithms can redact samples with a specific label $y$. In the data-based setting (Alg. 1), we express this as $\Omega = \{x \in X : \texttt{label}(x) = y\}$. In the validity-based setting (Alg. 2), we express this by setting $\mathbf{v}(x) = 1\{\arg\max_i \texttt{logit}(x)_i \neq y\}$, where $\texttt{logit}$ is the output of the softmax layer of a pre-trained label classifier [Chen, 2020]. In the classifier-based setting (Alg. 3), we set $\mathbf{f}(x) = 1 - \texttt{logit}(x)_y$.

In Table 1, we compare invalidity and generation quality among different algorithms and datasets when we redact label 0. We plot invalidity during data redaction in Fig. 2. We also compare invalidity after one epoch of data redaction in Appendix F.1.1. Mean and standard errors for 5 random runs are reported. Results for different hyper-parameters and redacting other labels are in Appendix F.

**Results.** We find all the algorithms in Section 3 work quite well with a much fewer number of epochs used for training the pre-trained model (which is 200). These algorithms are generally comparable. Therefore, we conclude that the simplest data-based algorithm is good enough to redact samples when those training samples to be redacted ($X \cap \Omega$) can characterize the redaction set ($\Omega$) well.

We also find invalidity rapidly drops after only one epoch of data redaction, indicating these algorithms are very efficient in penalizing invalidity. While different algorithms perform better on different datasets, they are highly comparable with each other. The reason why the classifier-based algorithm performs the best on MNIST is possibly that the label classifier on MNIST is almost perfect so its gradient information is accurate.

**Visualization.** We sample latents $z \sim \mathcal{N}(0, I)$ and choose those corresponding to invalid samples, i.e. $G_0(z) \in \Omega$ where $G_0$ is the pre-trained generator. We select visually good $G_0(z)$ for demonstration.

We visualize $G(z)$ during data redaction in Fig. 3, and more visualizations are in Appendix F.3. This demonstrates how the latent space is manipulated: the label to be redacted is gradually pushed to other labels, and there is high-level visual similarity between the final $G(z)$ and the original $G_0(z)$.

Table 2: Study on the effect of $T$ in Alg. 2 when the total number of queries is fixed. $R$ refers to the number of epochs of data redaction. A large $T$ may lead to worse invalidity.

| $T$ | MNIST | | | CIFAR-10 | | | STL-10 | | |
|---|---|---|---|---|---|---|---|---|---|
| | $R$ | $\texttt{Inv}(\downarrow)$ | $\texttt{IS}(\uparrow)$ | $R$ | $\texttt{Inv}(\downarrow)$ | $\texttt{FID}(\downarrow)$ | $R$ | $\texttt{Inv}(\downarrow)$ | $\texttt{FID}(\downarrow)$ |
| 400 | 20 | $\mathbf{0.0 \times 10^{-4}}$ | 7.10 | 75 | $\mathbf{0.45 \times 10^{-2}}$ | 35.1 | 100 | $1.0 \times 10^{-3}$ | $\mathbf{75.1}$ |
| 1000 | 8 | $0.6 \times 10^{-4}$ | $\mathbf{7.19}$ | 30 | $0.76 \times 10^{-2}$ | 34.8 | 40 | $\mathbf{0.8 \times 10^{-3}}$ | 77.0 |
| 2000 | 4 | $2.8 \times 10^{-4}$ | 7.11 | 15 | $1.00 \times 10^{-2}$ | $\mathbf{31.9}$ | 20 | $1.0 \times 10^{-3}$ | $\mathbf{75.1}$ |

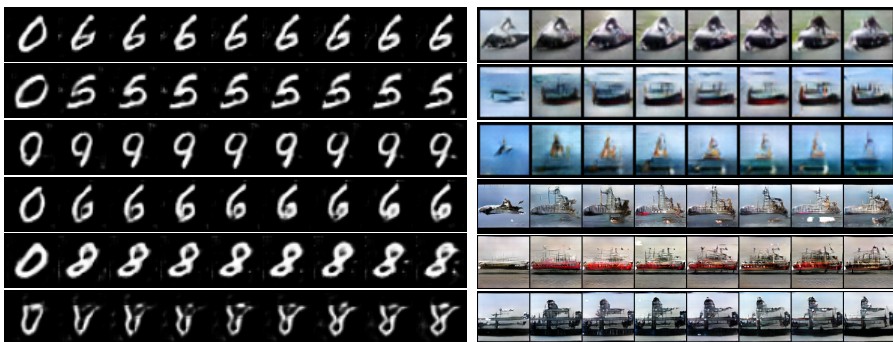

Figure 3: Visualization of the data redaction process of invalid samples when redacting label zero. The first column is generated by the pre-trained generator, and the $i$-th column is generated after $k \cdot (i-1)$ epochs of data redaction. Left: MNIST with $k = 1$. Right: top is CIFAR-10 and bottom is STL-10, both with $k = 4$ and label zero being `airplanes`. We can see samples associated with invalid labels are gradually pushed to other labels, but a high-level visual similarity is kept.

**Effects of other hyper-parameters.** In Table 2, we compare different $T$ (#queries after each epoch) in the validity-based redaction algorithm (Alg. 2). We fix the total number of queries by setting $T \times$#epochs to be a constant. Results indicate that a large $T$ may lead to worse invalidity, and there is trade-off between invalidity and quality when setting $T$ to be small or moderate.

In Appendix F.1.3, we compare different $\lambda$ (hyperparameter in (2)) in the classifier-based redaction algorithm (Alg. 3). We find there exists a clear trade-off between invalidity and quality when alternating $\lambda$: a larger $\lambda$ tends to produce better quality, and a smaller $\lambda$ tends to have better invalidity.

**Redacting multiple sets.** We then investigate how well the proposed algorithms can generalize to multiple redaction sets with methods in Section 3.4. We focus on the CelebA dataset [Liu et al., 2015], which has 40 labeled attributes. We use proposed algorithms to redact a combination of these attributes: $\Omega_1 = \{\texttt{Black\_hair and Blurry}\}$, $\Omega_2 = \{\texttt{Brown\_hair and Wear\_eyeglasses}\}$, and $\Omega = \Omega_1 \cup \Omega_2$. These attributes are randomly selected from those easy to capture. See detailed setup in Appendix F.4. Results after 1 or 5 epochs are reported in Table 3. Consistent with results on redacting just one label, all algorithms can reduce invalidity and retain generation quality and are comparable, while the classifier-based algorithm achieves the best invalidity after one epoch.

Table 3: Invalidity and generation quality of different redaction algorithms on redacting a combination of attributes within CelebA. There is a significant drop of invalidity, indicating that different redaction algorithms can all generalize to multiple redaction sets.

| Evaluation | Pre-trained | Epochs | Data-based | Data-based (sequentially) | Validity-based | Classifier-based |
|---|---|---|---|---|---|---|
| $\texttt{Inv}(\downarrow)$ | $1.66 \times 10^{-3}$ | 1 | $9.0 \times 10^{-4}$ | - | $7.6 \times 10^{-4}$ | $\mathbf{7.0 \times 10^{-4}}$ |
| $\texttt{Inv}(\downarrow)$ | $1.66 \times 10^{-3}$ | 5 | $\mathbf{3.8 \times 10^{-4}}$ | $6.0 \times 10^{-4}$ | $6.8 \times 10^{-4}$ | $6.8 \times 10^{-4}$ |
| $\texttt{FID}(\downarrow)$ | 36.4 | 5 | 29.3 | 28.6 | 29.9 | $\mathbf{27.9}$ |

### E.3 Model De-biasing

There can be different artifacts in GAN generated samples, and these could harm the overall generation quality. These artifacts may not exist in training samples, but are caused by inductive biases of the model, and become obvious after training. We can post-edit a pre-trained model to remove these artifacts, which we call *model de-biasing*. In this section, we investigate how well Alg. 2 and Alg. 3 apply to this task. We assume training samples are not biased so Alg. 1 does not apply to de-biasing.

To use these algorithms for de-biasing, we assume the target artifact or bias can be automatically detected by a classifier $\mathbf{f}$ or a validity function $\mathbf{v}$. Specifically, we survey two kinds of biases: boundary artifacts and label biases.

**Boundary artifacts.** A GAN trained on MNIST might generate samples that have numerous white pixels on the boundary (see Appendix G.1). We call this phenomenon the *boundary artifact*. We use the validity-based algorithm (Alg. 2) to de-bias boundary artifacts. The validity function is defined as $\mathbf{v}(x) = 1\{\sum_{(i,j)\in \text{boundary pixels}} x_{ij} < \tau_b\}$, where boundary pixels are those within a certain margin to the boundary, and threshold $\tau_b$ satisfies no training image is invalid.

Results are reported in Table 4. It is clear that the invalidity reduces in order after data redaction, indicating boundary artifacts are largely removed. Consistent with Table 2, a small or moderate $T$ leads to better results. We visualize samples before and after de-biasing in Appendix G.1.

Table 4: Invalidity after de-biasing boundary artifacts of generated MNIST samples. We run the validity-based redaction algorithm (Alg. 2) for 4 epochs. The invalidity drops significantly, and a small or moderate $T$ leads to slightly lower (better) invalidity.

|  | Pre-trained | $T = 5\text{K}$ | $T = 10\text{K}$ | $T = 20\text{K}$ | $T = 40\text{K}$ | $T = 80\text{K}$ |
|---|---|---|---|---|---|---|
| Margin $= 1$ | $3.1 \times 10^{-3}$ | $\mathbf{6.0 \times 10^{-5}}$ | $8.0 \times 10^{-5}$ | $2.0 \times 10^{-4}$ | $2.0 \times 10^{-4}$ | $7.0 \times 10^{-4}$ |
| Margin $= 2$ | $1.1 \times 10^{-3}$ | $1.6 \times 10^{-4}$ | $\mathbf{4.0 \times 10^{-5}}$ | $6.0 \times 10^{-5}$ | $3.2 \times 10^{-4}$ | $2.8 \times 10^{-4}$ |

Table 5: Invalidity and Inception scores after de-biasing label biases of generated samples from MNIST. We run the classifier-based redaction algorithm (Alg. 3) for 8 epochs with $\lambda = 0.8$, and compare to the data deletion baseline with 200 epochs of full re-training. The arrow means improvement from the pre-trained model to after data redaction. There is a clear improvement of generation quality, indicating the proposed algorithm can help GANs generate better samples. In contrast, data deletion does not help improve invalidity or quality.

| $\tau$ | Redaction (8 epochs) | | Data deletion baseline (200 epochs) | |
|---|---|---|---|---|
|  | $\texttt{Inv}(\downarrow)$ | $\texttt{IS}(\uparrow)$ | $\texttt{Inv}(\downarrow)$ | $\texttt{IS}(\uparrow)$ |
| 0.3 | $8.19 \times 10^{-4} \to 2.60 \times 10^{-4}$ | $7.82 \to \mathbf{8.10}$ | $8.19 \times 10^{-4} \to 1.14 \times 10^{-3}$ | $7.82 \to 7.75$ |
| 0.5 | $2.07 \times 10^{-2} \to 1.70 \times 10^{-2}$ | $7.82 \to 7.92$ | $2.07 \times 10^{-2} \to 2.17 \times 10^{-2}$ | $7.82 \to 7.79$ |
| 0.7 | $1.35 \times 10^{-1} \to 1.22 \times 10^{-1}$ | $7.82 \to 7.95$ | $1.35 \times 10^{-1} \to 1.32 \times 10^{-1}$ | $7.82 \to 7.82$ |

Table 6: Invalidity and FID scores after de-biasing label biases of generated samples from CIFAR-10. We run the classifier-based redaction algorithm (Alg. 3) for 30 epochs with $\lambda = 0.9$. The arrow means improvement from the pre-trained model to after data redaction. There is a clear improvement of generation quality, indicating the proposed algorithm can help GANs generate better samples. Note that there is **no** invalid sample in the training set, so the data deletion baseline is identical to the pre-trained model.

| $\tau$ | $\texttt{Inv}(\downarrow)$ | $\texttt{FID}(\downarrow)$ |
|---|---|---|
| 0.5 | $2.28 \times 10^{-2} \to 1.67 \times 10^{-2}$ | $36.2 \to \mathbf{26.6}$ |
| 0.7 | $1.72 \times 10^{-1} \to 1.49 \times 10^{-1}$ | $36.2 \to 26.8$ |
| 0.3 | $5.79 \times 10^{-4} \to 2.20 \times 10^{-4}$ | $36.2 \to 27.1$ |

**Label biases.** Neural networks may generate visually smooth but semantically ambiguous samples [Kirichenko et al., 2020], e.g. samples that look like multiple objects (see Appendix G.2). We call this phenomenon the *label bias*. We use the classifier-based algorithm (Alg. 3) to de-bias label

biases. The classifier is defined as $\mathbf{f}(x) = 1 - \text{Entropy}(\texttt{logit}(x))/\log(\texttt{\#classes})$, where the `logit` function is the same as in Appendix E.2. We also compare to a data deletion baseline, where we delete invalid samples and fully re-train the model. Results are reported in Table 5 and 6. After de-biasing, we can improve the generation quality by a significant gap ($\sim 0.3$ in IS and $\sim 10$ in FID). There is also a clear drop in terms of invalidity. In contrast, we find that data deletion does not help removing label biases.

## E.4    Understanding Training Data through the Lens of Data Redaction

Large datasets can be hard to analyze. In this section, we investigate how data redaction can help us understand these data. Specifically, we ask: which samples are easy or hard to redact?

In order to quantify the difficulty to redact a sample, we define the redaction score $\mathcal{FS}$ to be the difference of discriminator outputs before and after data redaction. Formally, let $x \in \Omega$ be a sample to redact, $\mathcal{M} = (G_0, D_0)$ be the pre-trained model, and $\mathcal{M}' = (G', D')$ be a model after data redaction. Then, the redaction score is $\mathcal{FS}(x) = D_0(x) - D'(x)$. A larger $\mathcal{FS}$ means it is easier to redact $x$.

To investigate **sample-level** redaction difficulty, we redact a particular label at one time using Alg. 1. We then demonstrate scatter plots of redaction scores $\mathcal{FS}(x)$ versus pre-trained discriminator outputs $D_0(x)$ for all samples $x$ with this label. We also fit linear regression and report $R^2$ values (larger means stronger linear relationship). Scatter plots for some labels in MNIST and CIFAR-100 and distribution of $R^2$ for all labels are shown in Fig. 4. We also visualize the most and least difficult-to-redact samples in Appendix H. We find there is positive correlation between $\mathcal{FS}(x)$ and $D_0(x)$, indicating on-manifold (large $D_0(x)$) samples are easier to be redacted, while off-manifold (small $D_0(x)$) ones are harder to be redacted. This analysis further provides a way to investigate **label-level** redaction difficulty. By averaging redaction scores for samples associated with each label, we can survey which labels are easy or hard to redact in general. The results are in Appendix H. We find some labels are harder to redact than others.

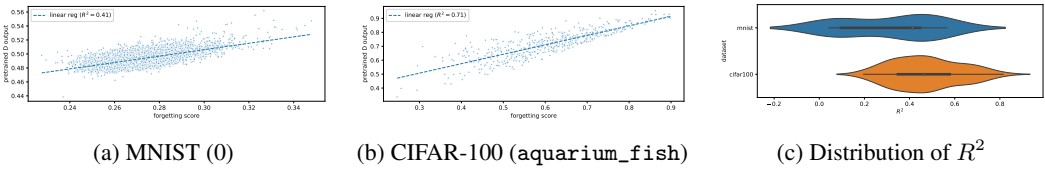

(a) MNIST (0)               (b) CIFAR-100 (`aquarium_fish`)          (c) Distribution of $R^2$

Figure 4: (a) and (b) Redaction scores of invalid training samples ($\mathcal{FS}(x)$) versus the pre-trained discriminator outputs of them ($D_0(x)$). There is positive correlation between these two scores, indicating on-manifold samples are easier to redact. (a) Redacting 0 in MNIST. (b) Redacting `aquarium_fish` in CIFAR-100. (c) Distributions of $R^2$ scores of linear regression between $\mathcal{FS}(x)$ and $D_0(x)$ for all labels. The correlation in (a) and (b) is universal and stronger in CIFAR-100.

## E.5    Discussion: Relationship to Adversarial Samples

An adversarial sample for a classifier $\mathbf{f}$ and a sample $x$ is another sample $\tilde{x} \approx x$ but $\mathbf{f}(\tilde{x}) \neq \mathbf{f}(x)$. Generating and defending these samples have become one of the most important directions of deep learning [Goodfellow et al., 2014b, Madry et al., 2018]. In this section, we show a variant of Alg. 3 can potentially be used to define a specific type of adversarial samples. In detail, we fix the discriminator $D$ and only update the generator $G$ while running Alg. 3. Then, $G$ is trained to fool $D$ and the classifier $\mathbf{f}$ at the same time. Notice that fooling $D$ means generating on-manifold (visually similar to training data) samples, and fooling $\mathbf{f}$ means finding adversarial samples of $\mathbf{f}$. By combining these objectives we can force $G$ to produce on-manifold adversarial samples, which may be significant in many real-world applications. We visualize some samples in Appendix I.

# F Redacting Labels

## F.1 Redacting Label 0

We include results for redacting label 0 in this section. We look at MNIST, CIFAR-10, and STL-10 datasets with different sets of hyper-parameters. With the base set of hyper-parameters, while different redaction algorithms perform better on different datasets, they are highly comparable with each other. We find the results are worse when there is no label smoothing ($\alpha_+ = 1, \alpha_- = 0$), indicating label smoothing is important for data redaction. We discuss results after one epoch in Appendix F.1.1, the effect of $\lambda$ in Appendix F.1.3, and the effect of $T$ in Table 2.

**Results for MNIST.**

Table 7: Data-based redaction algorithm.

| Model | Epochs | Inv($\downarrow$) | IS($\uparrow$) |
|---|---|---|---|
| pre-trained | 200 | $1.095 \times 10^{-1}$ | 7.82 |
| Base: $\alpha_+ = 0.95, \alpha_- = 0.05, \lambda = 0.85$ | 8 | 0 | 7.19 |
| $\alpha_+ = 0.9, \alpha_- = 0.1$ | 8 | $2 \times 10^{-5}$ | 7.02 |
| $\alpha_+ = 1.0, \alpha_- = 0.0$ | 8 | $4 \times 10^{-5}$ | 6.97 |
| $\lambda = 0.8$ | 8 | $2 \times 10^{-5}$ | 7.18 |
| $\lambda = 0.9$ | 8 | $4 \times 10^{-5}$ | 7.16 |
| $\lambda = 0.95$ | 8 | $5.2 \times 10^{-4}$ | 7.19 |
| $\lambda = 0.8$ | 1 | $2.98 \times 10^{-3}$ | 7.09 |

Table 8: Validity-based redaction algorithm.

| Model | Epochs | Inv($\downarrow$) | IS($\uparrow$) |
|---|---|---|---|
| pre-trained | 200 | $1.095 \times 10^{-1}$ | 7.82 |
| Base: $\alpha_+ = 0.95, \alpha_- = 0.05, \lambda = 0.85, T = 1000$ | 8 | $8 \times 10^{-5}$ | 7.17 |
| $\alpha_+ = 0.9, \alpha_- = 0.1$ | 8 | $3.4 \times 10^{-4}$ | 7.06 |
| $\alpha_+ = 1.0, \alpha_- = 0.0$ | 8 | $3.72 \times 10^{-3}$ | 4.81 |
| $\lambda = 0.8$ | 8 | 0 | 7.23 |
| $\lambda = 0.9$ | 8 | $2.2 \times 10^{-4}$ | 7.07 |
| $\lambda = 0.95$ | 8 | $8.8 \times 10^{-4}$ | 7.12 |
| $T = 400$ | 20 | 0 | 7.10 |
| $T = 2000$ | 4 | $2.8 \times 10^{-4}$ | 7.11 |
| $\lambda = 0.8$ | 1 | $2.80 \times 10^{-3}$ | 6.99 |

Table 9: Classifier-based redaction algorithm.

| Model | Epochs | Inv($\downarrow$) | IS($\uparrow$) |
|---|---|---|---|
| pre-trained | 200 | $1.095 \times 10^{-1}$ | 7.82 |
| Base: $\alpha_+ = 0.95, \alpha_- = 0.05, \lambda = 0.85, \tau = 0.5$ | 8 | $4 \times 10^{-5}$ | 7.19 |
| $\alpha_+ = 0.9, \alpha_- = 0.1$ | 8 | $1.4 \times 10^{-4}$ | 7.09 |
| $\alpha_+ = 1.0, \alpha_- = 0.0$ | 8 | $2.06 \times 10^{-3}$ | 6.08 |
| $\lambda = 0.8$ | 8 | $6 \times 10^{-5}$ | 7.15 |
| $\lambda = 0.9$ | 8 | $8 \times 10^{-5}$ | 7.18 |
| $\lambda = 0.95$ | 8 | $7.2 \times 10^{-4}$ | 7.24 |
| $\tau = 0.3$ | 8 | $1.2 \times 10^{-4}$ | 7.12 |
| $\tau = 0.7$ | 8 | $6 \times 10^{-5}$ | 7.22 |
| $\lambda = 0.8$ | 1 | $2.54 \times 10^{-3}$ | 7.11 |

**Results for CIFAR-10.**

Table 10: Data-based redaction algorithm.

| Model | Epochs | Inv($\downarrow$) | FID($\downarrow$) |
|---|---|---|---|
| pre-trained | 200 | $1.291 \times 10^{-1}$ | 36.2 |
| Base: $\alpha_+ = 0.9, \alpha_- = 0.05, \lambda = 0.8$ | 30 | $7.4 \times 10^{-3}$ | 35.8 |
| $\alpha_+ = 0.9, \alpha_- = 0.1$ | 30 | $8.0 \times 10^{-3}$ | 34.4 |
| $\alpha_+ = 0.9, \alpha_- = 0.0$ | 30 | $8.9 \times 10^{-3}$ | 34.2 |
| $\lambda = 0.9$ | 30 | $2.10 \times 10^{-2}$ | 29.2 |
| $\lambda = 0.95$ | 30 | $4.21 \times 10^{-2}$ | 26.2 |
| Base | 1 | $3.99 \times 10^{-2}$ | 37.1 |

Table 11: Validity-based redaction algorithm.

| Model | Epochs | Inv($\downarrow$) | FID($\downarrow$) |
|---|---|---|---|
| pre-trained | 200 | $1.291 \times 10^{-1}$ | 36.2 |
| Base: $\alpha_+ = 0.9, \alpha_- = 0.05, \lambda = 0.8, T = 1000$ | 30 | $7.9 \times 10^{-3}$ | 35.3 |
| $\alpha_+ = 0.9, \alpha_- = 0.1$ | 30 | $8.1 \times 10^{-3}$ | 33.8 |
| $\alpha_+ = 0.9, \alpha_- = 0.0$ | 30 | $8.1 \times 10^{-3}$ | 34.1 |
| $\lambda = 0.9$ | 30 | $2.54 \times 10^{-2}$ | 28.1 |
| $\lambda = 0.95$ | 30 | $3.57 \times 10^{-2}$ | 27.8 |
| $T = 400$ | 75 | $4.5 \times 10^{-3}$ | 35.1 |
| $T = 2000$ | 15 | $1.00 \times 10^{-2}$ | 31.9 |
| Base | 1 | $3.85 \times 10^{-2}$ | 36.2 |

Table 12: Classifier-based redaction algorithm.

| Model | Epochs | Inv($\downarrow$) | FID($\downarrow$) |
|---|---|---|---|
| pre-trained | 200 | $1.291 \times 10^{-1}$ | 36.2 |
| Base: $\alpha_+ = 0.9, \alpha_- = 0.05, \lambda = 0.8, \tau = 0.5$ | 30 | $1.28 \times 10^{-2}$ | 33.7 |
| $\alpha_+ = 0.9, \alpha_- = 0.1$ | 30 | $1.04 \times 10^{-2}$ | 32.9 |
| $\alpha_+ = 0.9, \alpha_- = 0.0$ | 30 | $6.3 \times 10^{-3}$ | 32.8 |
| $\lambda = 0.9$ | 30 | $2.25 \times 10^{-2}$ | 28.6 |
| $\lambda = 0.95$ | 30 | $4.26 \times 10^{-2}$ | 26.9 |
| $\tau = 0.3$ | 30 | $9.6 \times 10^{-3}$ | 34.8 |
| $\tau = 0.7$ | 30 | $1.05 \times 10^{-2}$ | 35.2 |
| Base | 1 | $3.47 \times 10^{-2}$ | 37.8 |

**Results for STL-10.**

Table 13: Data-based redaction algorithm.

| Model | Epochs | Inv($\downarrow$) | FID($\downarrow$) |
|---|---|---|---|
| pre-trained | 200 | $6.23 \times 10^{-2}$ | 79.1 |
| Base: $\alpha_+ = 0.9, \alpha_- = 0.05, \lambda = 0.8$ | 40 | $7.8 \times 10^{-4}$ | 74.3 |
| $\alpha_+ = 0.9, \alpha_- = 0.1$ | 40 | $7.6 \times 10^{-4}$ | 75.8 |
| $\alpha_+ = 0.9, \alpha_- = 0.0$ | 40 | $1.42 \times 10^{-3}$ | 82.7 |
| $\lambda = 0.9$ | 40 | $2.88 \times 10^{-3}$ | 76.9 |
| $\lambda = 0.95$ | 40 | $6.71 \times 10^{-3}$ | 78.2 |
| Base | 1 | $6.97 \times 10^{-3}$ | 75.1 |

Table 14: Validity-based redaction algorithm.

| Model | Epochs | Inv($\downarrow$) | FID($\downarrow$) |
|---|---|---|---|
| pre-trained | 200 | $6.23 \times 10^{-2}$ | 79.1 |
| Base: $\alpha_+ = 0.9, \alpha_- = 0.05, \lambda = 0.8, T = 1000$ | 40 | $4.8 \times 10^{-4}$ | 79.3 |
| $\alpha_+ = 0.9, \alpha_- = 0.1$ | 40 | $8.2 \times 10^{-4}$ | 76.5 |
| $\alpha_+ = 0.9, \alpha_- = 0.0$ | 40 | $1.44 \times 10^{-3}$ | 77.0 |
| $\lambda = 0.9$ | 40 | $4.52 \times 10^{-3}$ | 75.9 |
| $\lambda = 0.95$ | 40 | $8.95 \times 10^{-3}$ | 75.3 |
| $T = 400$ | 100 | $1.00 \times 10^{-3}$ | 75.1 |
| $T = 2000$ | 20 | $1.00 \times 10^{-3}$ | 75.1 |
| Base | 1 | $8.99 \times 10^{-3}$ | 79.5 |

Table 15: Classifier-based redaction algorithm.

| Model | Epochs | Inv($\downarrow$) | FID($\downarrow$) |
|---|---|---|---|
| pre-trained | 200 | $6.23 \times 10^{-2}$ | 79.1 |
| Base: $\alpha_+ = 0.9, \alpha_- = 0.05, \lambda = 0.8, \tau = 0.5$ | 40 | $8.6 \times 10^{-4}$ | 75.4 |
| $\alpha_+ = 0.9, \alpha_- = 0.1$ | 40 | $9.2 \times 10^{-4}$ | 74.8 |
| $\alpha_+ = 0.9, \alpha_- = 0.0$ | 40 | $1.62 \times 10^{-3}$ | 82.0 |
| $\lambda = 0.9$ | 40 | $3.10 \times 10^{-3}$ | 77.2 |
| $\lambda = 0.95$ | 40 | $6.89 \times 10^{-3}$ | 76.2 |
| $\tau = 0.3$ | 40 | $8.8 \times 10^{-4}$ | 73.8 |
| $\tau = 0.7$ | 40 | $1.34 \times 10^{-3}$ | 76.1 |
| Base | 1 | $6.81 \times 10^{-3}$ | 75.6 |

### F.1.1 Invalidity after one epoch.

We compare invalidity after only one epoch of data redaction. These redaction algorithms are highly comparable to each other. We hypothesis that the classifier-based algorithm performs the best on MNIST because a label classifier on MNIST (and its gradient information) can be very accurate, while this may not be true for CIFAR-10 and STL-10.

Table 16: Invalidity after one epoch of data redaction.

| Dataset | Scale | Pre-trained | Data-based | Validity-based | Classifier-based |
|---------|-------|-------------|------------|----------------|------------------|
| MNIST | $\times 10^{-3}$ | $1.1 \times 10^2$ | $4.7 \pm 0.8$ | $5.6 \pm 0.9$ | $\mathbf{3.9 \pm 0.9}$ |
| CIFAR-10 | $\times 10^{-2}$ | $1.3 \times 10^1$ | $\mathbf{3.7 \pm 0.5}$ | $3.7 \pm 0.8$ | $3.8 \pm 0.3$ |
| STL-10 | $\times 10^{-3}$ | $6.2 \times 10^1$ | $9.1 \pm 0.9$ | $\mathbf{8.6 \pm 0.9}$ | $10.6 \pm 1.2$ |

### F.1.2 Quality during data redaction

We plot quality measure of different data redaction algorithms on different datasets during the redaction process, complementary to the invalidity in Fig. 2. We find the variances of quality measure is higher than the invalidity, but different redaction algorithms are generally comparable.

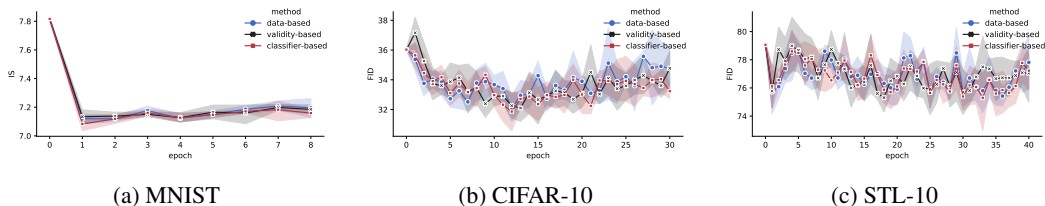

| (a) MNIST | (b) CIFAR-10 | (c) STL-10 |

Figure 5: Quality measure during data redaction. Mean and standard errors are plotted for five random seeds.

### F.1.3 Trade-off by alternating $\lambda$.

We study the effect of $\lambda$ (hyper-parameter in (2)) in Table 17 and Fig. 6. There is a trade-off by alternating $\lambda$: a larger $\lambda$ (less fake data from the redaction set) leads to better quality measure, and a smaller $\lambda$ (more fake data from the redaction set) leads to better invalidity.

Table 17: Invalidity after data redaction for different $\lambda$ in the classifier-based redaction algorithm.

| $\lambda$ | MNIST | | CIFAR-10 | | STL-10 | |
|-----------|-------|-------|----------|-------|--------|-------|
| | Inv($\downarrow$) | IS($\uparrow$) | Inv($\downarrow$) | FID($\downarrow$) | Inv($\downarrow$) | FID($\downarrow$) |
| 0.8 | $0.6 \times 10^{-4}$ | 7.15 | $1.28 \times 10^{-2}$ | 33.7 | $0.86 \times 10^{-3}$ | 75.4 |
| 0.9 | $0.8 \times 10^{-4}$ | 7.18 | $2.25 \times 10^{-2}$ | 28.6 | $3.10 \times 10^{-3}$ | 77.2 |
| 0.95 | $7.2 \times 10^{-4}$ | 7.24 | $4.26 \times 10^{-2}$ | 26.9 | $6.89 \times 10^{-3}$ | 76.2 |

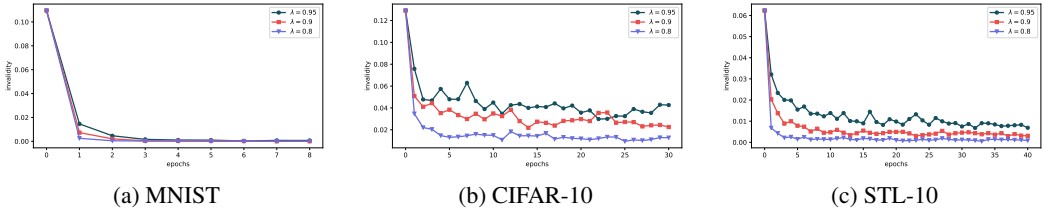

| (a) MNIST | (b) CIFAR-10 | (c) STL-10 |

Figure 6: Invalidity during data redaction for different $\lambda$ in the classifier-based redaction algorithm.

## F.2 Redacting Other Labels

We also demonstrate results for redacting other labels with our data redaction algorithms. We use the base set of hyper-parameters in Appendix F.1. Similar to redacting label 0, all redaction algorithms can largely reduce invalidity, and they are highly comparable to each other. The classifier-based redaction algorithm achieves slightly better generation quality on MNIST and CIFAR-10. In terms of different labels, we find some labels are harder to redact in the sense that the invalidity scores for these labels are higher than other scores, such as label 9 in MNIST, and label 3 in CIFAR-10 and STL-10.

Table 18: Redacting other labels on MNIST.

| Label | Pre-trained | | Data-based | | Validity-based | | Classifier-based | |
|---|---|---|---|---|---|---|---|---|
| | Inv($\downarrow$) | IS($\uparrow$) | Inv($\downarrow$) | IS($\uparrow$) | Inv($\downarrow$) | IS($\uparrow$) | Inv($\downarrow$) | IS($\uparrow$) |
| 1 | 10.2% | 7.81 | 0.002% | 7.01 | **0.000**% | **7.21** | 0.008% | 7.13 |
| 2 | 8.6% | 7.81 | 0.022% | 7.22 | **0.012**% | 7.20 | 0.028% | **7.28** |
| 3 | 11.5% | 7.81 | **0.126**% | 7.20 | 0.136% | **7.24** | 0.134% | 7.19 |
| 4 | 9.9% | 7.81 | 0.138% | 7.19 | **0.092**% | 7.21 | 0.104% | **7.26** |
| 5 | 8.7% | 7.81 | 0.048% | 7.22 | **0.046**% | 7.21 | 0.056% | **7.24** |
| 6 | 9.0% | 7.81 | 0.020% | 7.04 | 0.022% | 7.07 | **0.010**% | **7.12** |
| 7 | 11.4% | 7.81 | 0.114% | 7.24 | 0.124% | **7.34** | **0.088**% | 7.32 |
| 8 | 9.1% | 7.81 | **0.198**% | 7.48 | 0.248% | 7.35 | 0.302% | **7.51** |
| 9 | 10.7% | 7.81 | 0.486% | 7.30 | **0.414**% | **7.36** | 0.545% | 7.26 |

Table 19: Redacting other labels on CIFAR-10.

| Label | Pre-trained | | Data-based | | Validity-based | | Classifier-based | |
|---|---|---|---|---|---|---|---|---|
| | Inv($\downarrow$) | FID($\downarrow$) | Inv($\downarrow$) | FID($\downarrow$) | Inv($\downarrow$) | FID($\downarrow$) | Inv($\downarrow$) | FID($\downarrow$) |
| 1 | 1.5% | 36.24 | 0.032% | 35.06 | **0.014**% | 35.23 | 0.082% | **33.40** |
| 2 | 11.0% | 36.24 | **1.311**% | 31.67 | 1.537% | 31.65 | 1.564% | **28.34** |
| 3 | 15.8% | 36.24 | 3.013% | 30.10 | 3.491% | 31.01 | **2.534**% | **28.06** |
| 4 | 16.8% | 36.24 | 1.752% | 30.36 | 1.754% | 31.26 | **1.590**% | **29.72** |
| 5 | 6.7% | 36.24 | **0.799**% | **30.76** | 0.985% | 30.90 | 1.461% | 31.36 |
| 6 | 9.3% | 36.24 | 0.797% | 29.81 | 1.071% | 31.65 | **0.755**% | **29.64** |
| 7 | 8.6% | 36.24 | 0.789% | 33.48 | **0.496**% | **33.40** | 1.325% | 34.15 |
| 8 | 10.3% | 36.24 | **0.218**% | 38.96 | 1.451% | 38.59 | 0.496% | **34.56** |
| 9 | 7.1% | 36.24 | **0.138**% | 38.13 | 0.186% | 37.74 | 0.216% | **36.85** |

Table 20: Redacting other labels on STL-10.

| Label | Pre-trained | | Data-based | | Validity-based | | Classifier-based | |
|---|---|---|---|---|---|---|---|---|
| | Inv($\downarrow$) | FID($\downarrow$) | Inv($\downarrow$) | FID($\downarrow$) | Inv($\downarrow$) | FID($\downarrow$) | Inv($\downarrow$) | FID($\downarrow$) |
| 1 | 9.0% | 79.00 | **1.273**% | 74.89 | 2.168% | **73.91** | 1.900% | 75.34 |
| 2 | 6.2% | 79.00 | 0.158% | **72.22** | **0.132**% | 72.39 | 0.176% | 75.75 |
| 3 | 14.9% | 79.00 | 3.772% | 77.24 | **3.732**% | 76.80 | 4.412% | **75.19** |
| 4 | 8.2% | 79.00 | 1.634% | **81.91** | **1.345**% | 82.82 | 1.425% | 83.25 |
| 5 | 15.1% | 79.00 | **2.072**% | **76.85** | 3.383% | 80.40 | 5.041% | 77.74 |
| 6 | 8.7% | 79.00 | **0.462**% | 80.82 | 0.518% | **78.17** | 0.745% | 79.63 |
| 7 | 10.7% | 79.00 | 2.973% | 77.53 | **1.838**% | 78.57 | 2.180% | 77.58 |
| 8 | 9.5% | 79.00 | 0.304% | 79.56 | **0.272**% | 78.06 | 0.352% | **77.07** |
| 9 | 11.6% | 79.00 | **0.817**% | 76.70 | 0.947% | 78.37 | 0.941% | **76.37** |

## F.3 Visualization

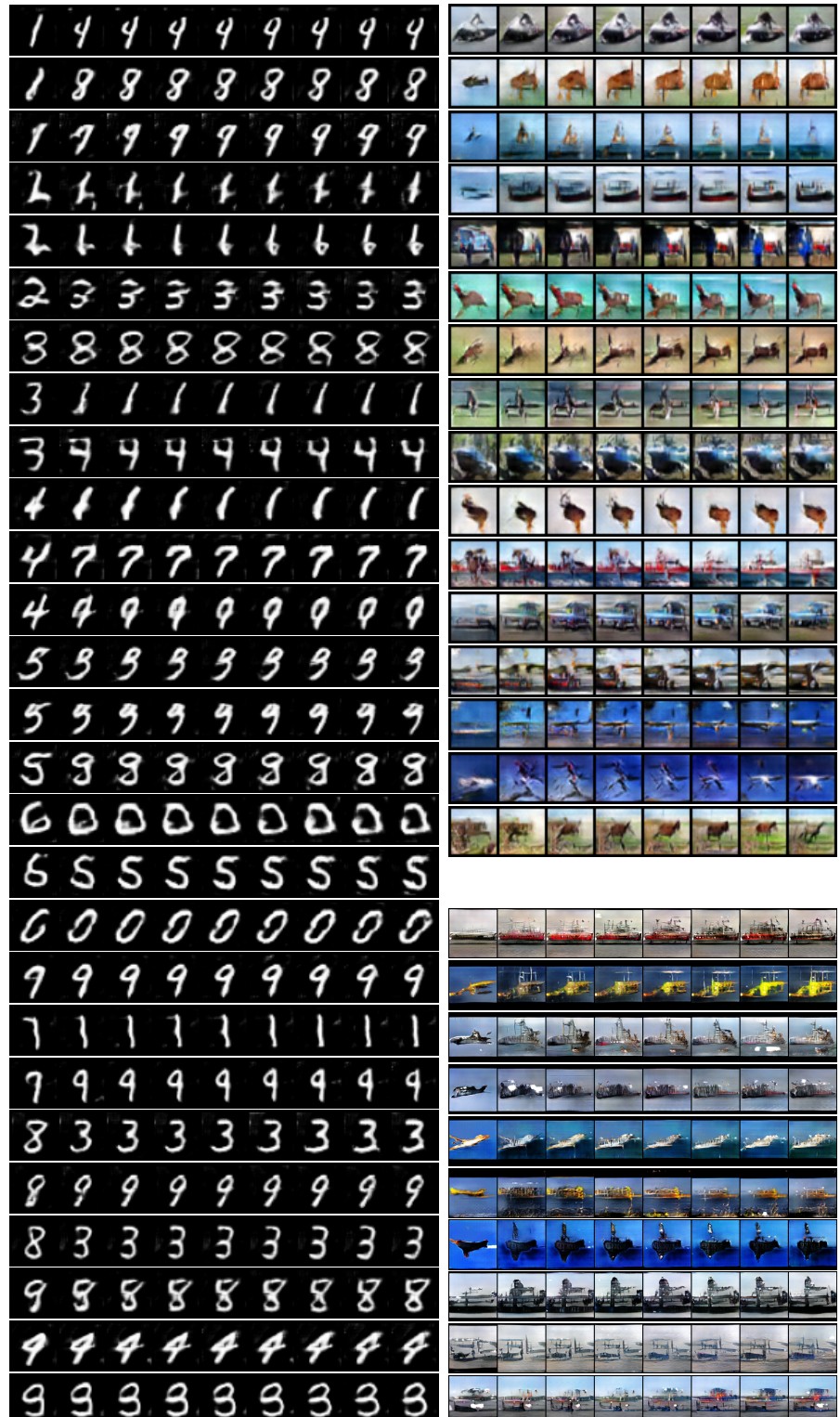

Figure 7: Visualization of the data redaction process of invalid samples when redacting labels. The first column is generated by the pre-trained generator, and the $i$-th column is generated after $k \cdot (i-1)$ epochs of data redaction. Left: MNIST with $k = 1$. Right: top is CIFAR-10 and bottom is STL-10, both with $k = 4$. We can see samples associated with invalid labels are gradually pushed to other labels, but a high-level visual similarity is kept.

## F.4 Detailed Setup of Redacting Multiple Sets

We use 30K images from CelebA-64 as the training set. All other hyper-parameters are the same as the base set for STL-10 in Appendix F.1, except that we run data redaction algorithms for only 5 epochs. We train attribute classifiers for each attribute separately. The attribute classifiers are fine-tuned from open-sourced pre-trained ResNet [He et al., 2016]. [4] We fine-tune the network for 20 epochs using the SGD optimizer with learning rate $= 1 \times 10^{-3}$, momentum $= 0.9$, and a batch size of 64.

---

[4]https://pytorch.org/vision/stable/models.html

# G  Model De-biasing

## G.1  Boundary Artifacts

Let the image size be $W \times H$ (the number of channels is 1 for MNIST). For an integer margin, the boundary pixels are defined as

$$\{(i,j) : 1 \leq i \leq \text{margin or } W - \text{margin} < i \leq W, 1 \leq j \leq \text{margin or } H - \text{margin} < j \leq H\}.$$

Then, the validity function for boundary artifacts is defined as

$$\mathbf{v}(x) = 1 \left\{ \sum_{(i,j) \in \text{boundary pixels}} x_{ij} < \tau_b \right\},$$

where $\tau_b = 4.25$ for $\text{margin} = 1$ and $10.0$ for $\text{margin} = 2$. For these values, no training data has the boundary artifact. Quantitative results are in Table 4. We visualize some samples with boundary artifacts in Fig. 8a. We run the validity-based redaction algorithm with $\lambda = 0.98, \alpha_+ = 0.95, \alpha_- = 0.05$ for 4 epochs. After de-biasing via data redaction, these samples have less boundary pixels, as shown in Fig. 8b.

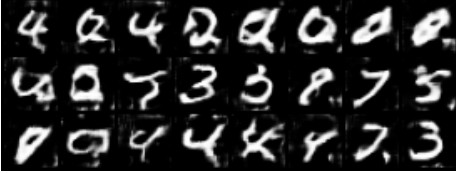

(a) Samples with boundary artifacts.      (b) Samples after de-biasing via data redaction.

Figure 8: De-biasing boundary artifacts with the validity-based data redaction algorithm. Margin $= 1$ and $T = 40$K.

## G.2  Label Biases

We use classifier-based redaction algorithm to de-bias label biases. For MNIST, we use $\lambda = 0.8, \alpha_+ = 0.95, \alpha_- = 0.05$ and run for 8 epochs. For CIFAR-10, we use $\lambda = 0.0, \alpha_+ = 0.9, \alpha_- = 0.05$ and run for 30 epochs. We visualize semantically ambiguous samples generated by the pre-trained model in Fig. 9a. After de-biasing via data redaction, these samples become less semantically ambiguous, as shown in Fig. 9b.

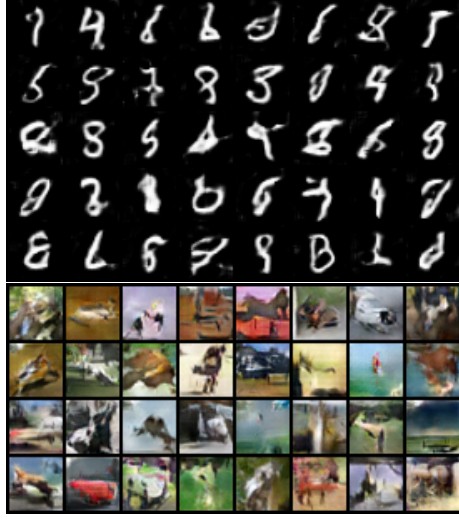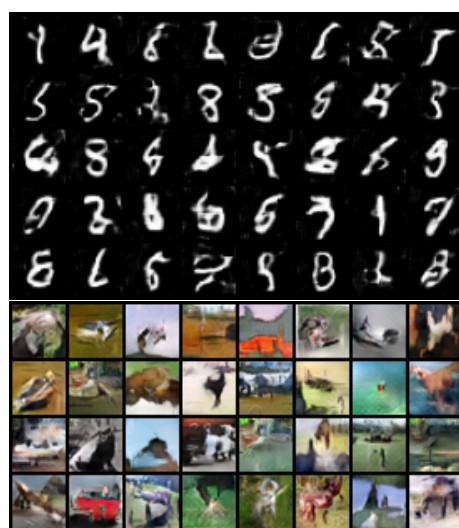

(a) Samples with label-biases.      (b) Samples after de-biasing via data redaction.

Figure 9: De-biasing label biases with the classifier-based data redaction algorithm ($\tau = 0.7$).

# H    Understanding Training Data

**Sample-level redaction difficulty.** We visualize some most and least difficult-to-redact samples according to the redaction scores in Fig. 10. We find the most difficult-to-redact samples are visually atypical, while the least difficult-to-redact samples are visually more common.

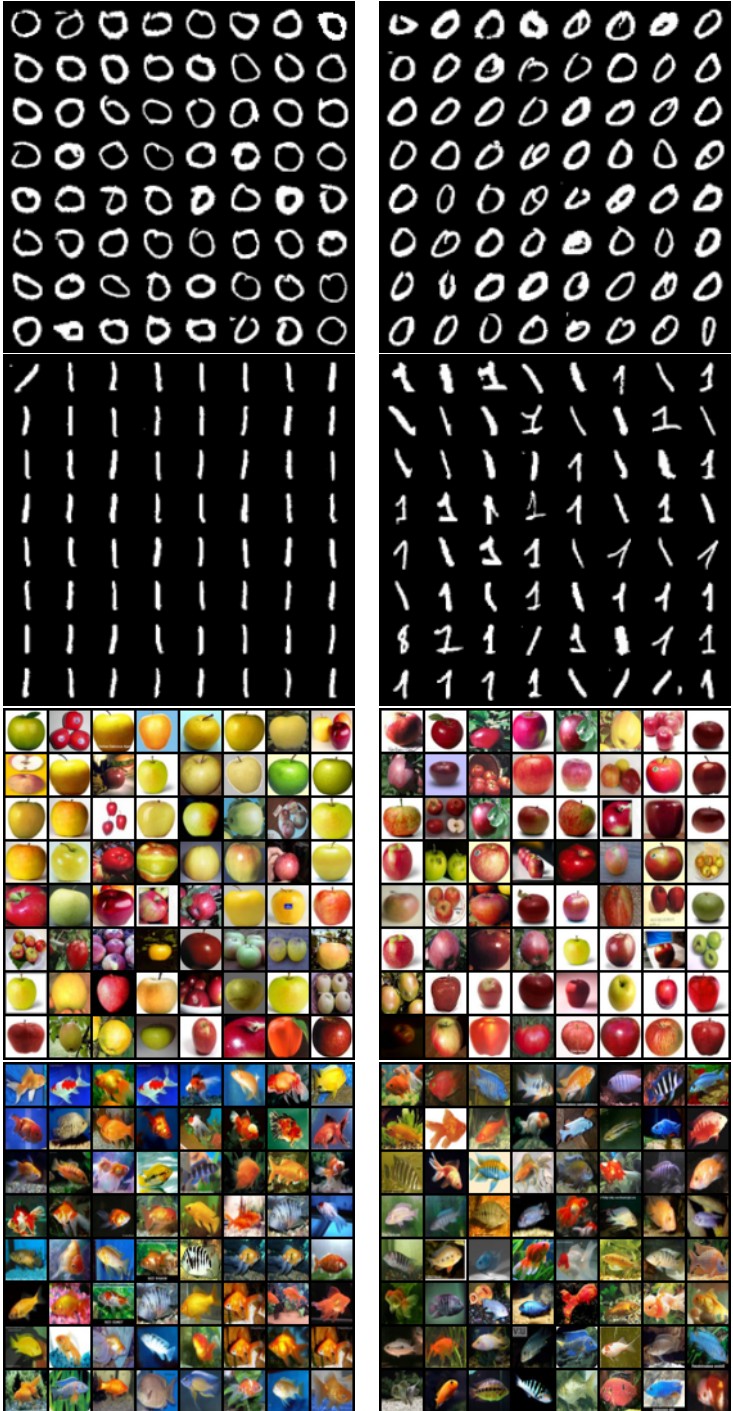

(a) Samples that are easiest to redact.   (b) Samples that are hardest to redact.

Figure 10: Samples that are most and least difficult-to-redact in MNIST and CIFAR-100.

**Label-level redaction difficulty.** We sort all labels according to their average redaction scores. This tells us which labels are easier or harder to redact. The results for MNIST are in Fig. 11a. Consistent with Table 18, label 9 is the most difficult label to redact. The least and most difficult-to-redact labels for CIFAR-100 are shown in Fig. 11c and 11b.

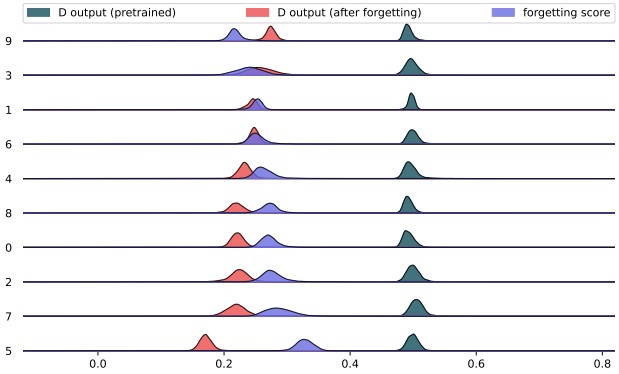

(a) Label-level redaction difficulty for MNIST. Top: the most difficult to redact. Bottom: the least difficult to redact.

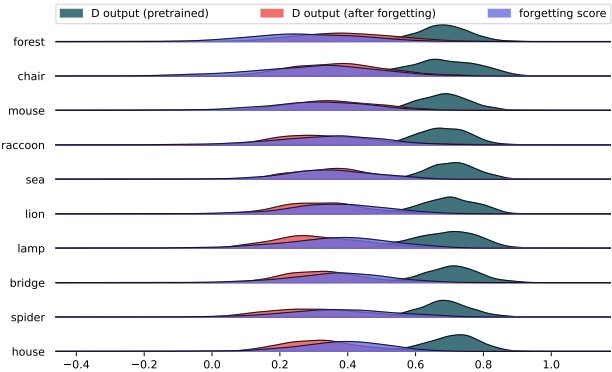

(b) Label-level redaction difficulty for CIFAR-100 (10 most difficult-to-redact labels). Top: the most difficult to redact. Bottom: the least difficult to redact.

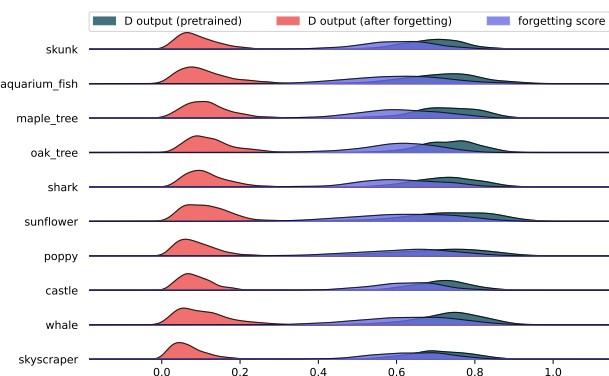

(c) Label-level redaction difficulty for CIFAR-100 (10 least difficult-to-redact labels). Top: the most difficult to redact. Bottom: the least difficult to redact.

Figure 11: Label-level redaction difficulty for MNIST and CIFAR-100. A large redaction score means a label is easier to be redacted. We find some labels are more difficult to redact than others.

# I  Relationship to Adversarial Samples

Consider the classifier-based redaction algorithm. We fix the discriminator and only update the generator. Then, the generator is trained to fool both the discriminator and the classifier at the same time. We may define generated samples from this generator as on-manifold adversarial samples to $(D, \mathbf{f})$. Note that on-manifold samples are not necessarily visually clear samples; instead, they could be high likelihood samples according to the inductive bias of the generative model.

When training, we use the classifier-based redaction algorithm with $\tau = 0.5, \alpha_+ = 0.95, \alpha_- = 0.05, \lambda = 0.85$, and a batch size of 64. We "redact" one label at a time, similar to experiments in Appendix F.2. After each iteration over one mini-batch, we generate samples with the same latents. The visualization is shown below. There are several interesting findings shown in the figures. First, the generated samples tend to have less pixels. Second, the generated samples tend to be dis-connected. Third, there are some general patterns across these generated samples (for each label): for example, there are pixels in the middle of zeroes, the bottom of sevens vanish, and nines are split from the middle. We conjecture that these observations correspond to the inductive bias of the discriminator and adversarial samples of the classifier.

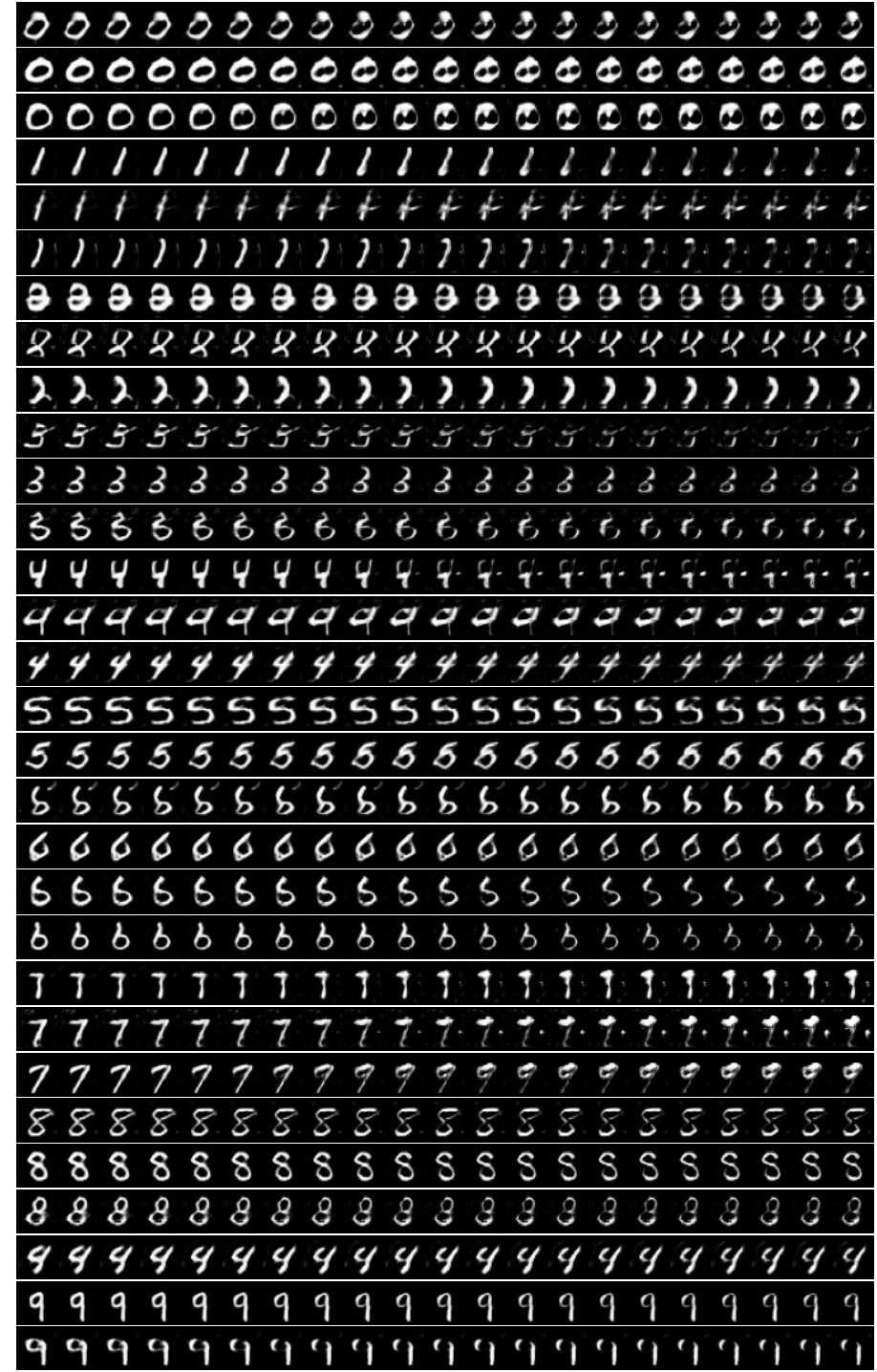

Figure 12: Generated samples when we only train the generator and fix the discriminator and the classifier with the classifier-based redaction algorithm. The first column is generated by the pre-trained generator, and the $i$-th column is generated after $i-1$ iterations (up to 20 iterations).

