# OpenReview forum: "Data Redaction from Pre-trained GANs"
_NeurIPS.cc/2022/Workshop/TSRML — TSRML2022_

### Official Review · Reviewer_cLnZ · 2022-10-17
**A well-written paper with limited novelty**

**Overall Rating:** 6

**Summary:**

In this paper, the author takes a compute-friendly approach and investigates how to post-edit a model after training so that it “forgets”, or refrains from outputting certain kinds of samples. Specifically, they define the data forgetting problem and then explain the difference between data deletion and data forgetting. Finally, the paper provides an approach by training with a custom-defined loss function.

**Strengths:**

1. The paper is well-written. The author gives a very clear comparison between data deletion and data forgetting.

2. From the experiment results, the provided method successfully reduces the generation distribution mass on the forgetting set.

**Weaknesses:**

1. The author may consider comparing the provided method with some data deletion methods on the invalidity metrics as a baseline.

2. The provided method is naive and lacks novelty.

**Overall Recommendation:**

Overall, I think this paper is a good paper. The author raises a problem, gives a clear definition and analysis of this problem, provides a solution, and did some experiments to show the results. But I think the provided method is naive and lacks novelty. The data forgetting problem is not a well-known problem, so the application of this method would be limited.

**Review Confidence:**

4: The reviewer is confident but not absolutely certain that the evaluation is correct

---

### Official Review · Reviewer_7AX2 · 2022-10-20
**unlearning GAN**

**Overall Recommendation:** Reject. see weakness.
**Overall Rating:** 3

**Summary:**

A paper about how to do machine unlearning of GAN, without the need of full re-training.

**Strengths:**

1.  The paper is well written but seems not to prepare for this workshop at all.
2. Detailed description of the method.

**Weaknesses:**

1. It seems that there is no experiment in section 4?
2. The paper seems to be a rush version for workshop.

**Review Confidence:**

5: The reviewer is absolutely certain that the evaluation is correct and very familiar with the relevant literature

---

### Official Review · Reviewer_znFJ · 2022-10-21

**Overall Rating:** 7

**Summary:**

This paper presents algorithms to post-edit a generative adversarial network (GAN) such that the model "forgets", i.e., does not generate samples with certain properties. The authors consider three types of description for the "properties": data-based, validity-based, and classifier-based, and propose corresponding algorithms targeting at the three descriptions. Essentially, they treat data with unwanted properties as fake data, such that the generators learn to avoid generating them. The authors provide extensive evaluations to understand several practical problems.



**Strengths:**

1. The paper formalizes the problem of data forgetting in accordance to several practical needs and clarifies its difference from data deletion.
2. The paper rigorously provides three types of descriptions of the forgetting sets and proposes corresponding algorithms targeting at each description.
3. The paper provides extensive evaluations to understand several relevant questions.
4. The paper is well-written and overall easy to understand.

**Weaknesses:**

1. **Data deletion vs. data forgetting**: I appreciate it that the authors try to clarify the differences between data deletion and data forgetting, but the two are *innately different* from their motivation (and therefore the goal to achieve). The aim of data deletion is privacy protection, i.e., removing the influence of the selected datapoints on the outcome of the algorithm; the aim of data forgetting is to explicitly regulate the algorithm output. I don't think the two are comparable. And it makes little sense to compare data forgetting algorithms with data deletion algorithms on data forgetting tasks---data deletion algorithms are not designed for this purpose at all.
2. **Validity-based vs. classifier-based**: My understanding for the difference between the validity-based forgetting set and the classifier-based forgetting set is that the validity function $\mathbf{v}$ may not necessarily be differentiable. And the classifier-based one is essentially a special case of the validity-based one (when the validity function happens to be differentiable). I thus doubt line 202 "The classifier-based $\Omega$ generalizes the validity-based $\Omega$." is a correct expression. Moreover, the logic in this paragraph is weird. It is exactly the differentiable property for this special case that gives rise to the "potentially useful information such as values and gradients". The algorithm leveraging these will naturally have better results. Framing this logic as "we can use a more general approach to solve the problem, but the more general approach will not give as good result as the specific approach" sounds unnatural.
3. In lines 205-206, the authors said "we will evaluate this effect in experiments". I hope the authors can specify where exactly in Appendix did they provide the evaluation.
4. **Experiments**: Sec. 4 basically describes the problems to answer by evaluation, but didn't provide the corresponding answers/conclusions. The readers wouldn't gain adequate understandings without referring to the Appendix. I would recommend the authors to provide brief conclusions in the main body so it is self-contained.
5. **Extension to other generative models**: In this paper, the authors restrict the discussion to one specific type of generative models, GAN. I'd like to see a discussion on its extension to broader types of generative models.

**Minors**:

* Typos
  * Line 2: "contributes to their trustworthiness" --> "contributes to their untrustworthiness"
  * Line 40: "after deploymen" --> "after deployment"
  * Line 181: "We hypothesis" --> "We hypothesize"
  * Line 233: "limitation or" --> "limitation of"
* Line 148: the symbol # is undefined
* Lines 39-40: "Classifiers can take a significant amount of space and time after deploymen" I don't understand the sentence. It is either "classifiers can take a significant amount of space", or "classification can take a significant amount of time".
* In lines 53-54, the authors said "we need to carefully balance data forgetting with retaining good generation quality". But I don't see why the two needs to be balanced. From what I see in the paper, the goal of "data forgetting" is mainly to avoid the artifacts in the generation, which naturally leads to better quality of the generation. It is true that data deletion needs a balance, but in practical use cases in data forgetting, the need to balance is less straightforward.


**Overall Recommendation:**

The paper formalizes the problem of data forgetting, comprehensively studies several possible scenarios (descriptions), and provides extensive evaluation on practical use cases. There exist a few flaws in the argument and writing, but overall it's a nice paper to read. I thus recommend acceptance.

**Review Confidence:**

3: The reviewer is fairly confident that the evaluation is correct

---

### Decision · Program_Chairs · 2022-10-23

**Decision:**

Accept

**Comment:**

The paper studies an interesting new problem of forgetting data in generative models. I feel the paper is a good fit to the workshop, however the authors should make sure to improve the structure and writing in the final version. The current version looks rush and not self-contained.